# Improving Text-to-Image Consistency via Automatic Prompt Optimization

**Oscar Mañas**[*]                                                                oscar.manas@mila.quebec
*Mila, Université de Montréal, Meta FAIR*

**Pietro Astolfi**[*]
*Meta FAIR*

**Melissa Hall**
*Meta FAIR*

**Candace Ross**
*Meta FAIR*

**Jack Urbanek**
*Meta FAIR*

**Adina Williams**
*Meta FAIR*

**Aishwarya Agrawal**
*Mila, Université de Montréal, Canada CIFAR AI Chair*

**Adriana Romero-Soriano**
*Mila, McGill University, Meta FAIR, Canada CIFAR AI Chair*

**Michal Drozdzal**
*Meta FAIR*

[*]*Contributed equally*

**Reviewed on OpenReview:** *https://openreview.net/forum?id=g12Gdl6aDL*

## Abstract

Impressive advances in text-to-image (T2I) generative models have yielded a plethora of high performing models which are able to generate aesthetically appealing, photorealistic images. Despite the progress, these models still struggle to produce images that are consistent with the input prompt, oftentimes failing to capture object quantities, relations and attributes properly. Existing solutions to improve prompt-image consistency suffer from the following challenges: (1) they oftentimes require model fine-tuning, (2) they only focus on nearby prompt samples, and (3) they are affected by unfavorable trade-offs among image quality, representation diversity, and prompt-image consistency. In this paper, we address these challenges and introduce a T2I optimization-by-prompting framework, OPT2I, which leverages a large language model (LLM) to improve prompt-image consistency in T2I models. Our framework starts from a user prompt and iteratively generates revised prompts with the goal of maximizing a consistency score. Our extensive validation on two datasets, MSCOCO and PartiPrompts, shows that OPT2I can boost the initial consistency score by up to 24.9% in terms of DSG score while preserving the FID and increasing the recall between generated and real data. Our work paves the way toward building more reliable and robust T2I systems by harnessing the power of LLMs.

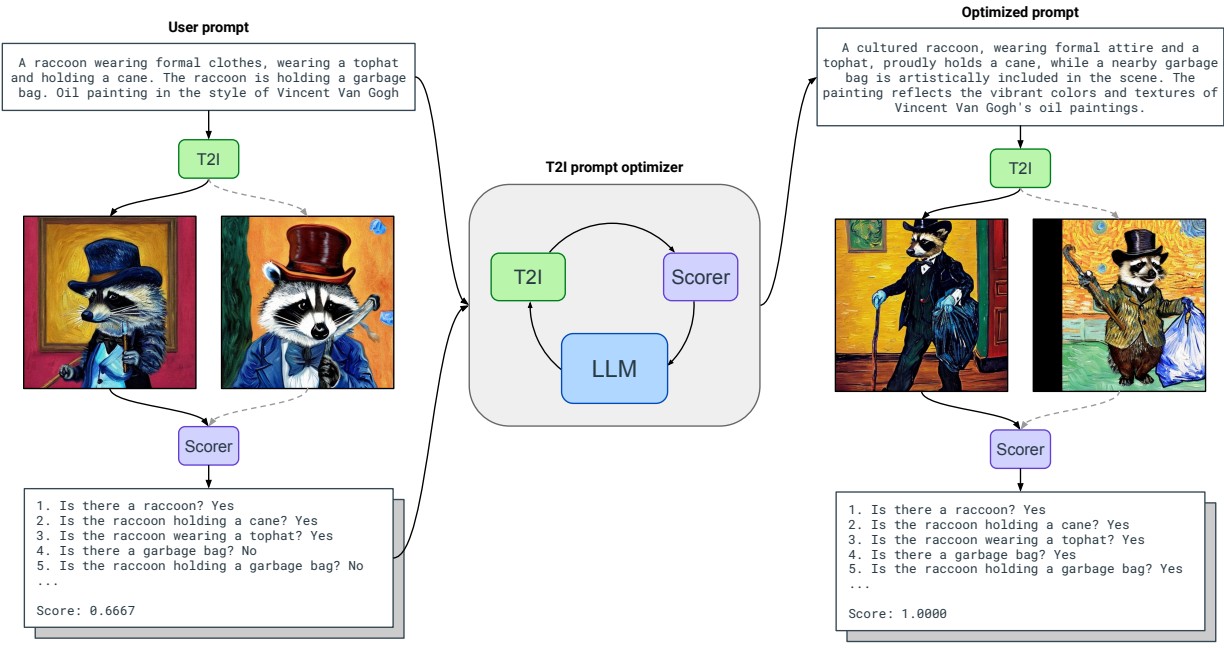

Figure 1: Overview of our backpropagation-free text-to-image optimization by prompting approach that rewrites user prompts with the goal of improving prompt-image consistency. Our framework is composed of a text-to-image generative model (T2I), a large language model (LLM) and a consistency objective (Scorer). The LLM iteratively leverages a history of prompt-score pairs to suggest revised prompts. In the depicted example, our system improves the consistency score by over 30% in terms of Davidsonian Scene Graph score.

## 1  Introduction

In recent years, we have witnessed remarkable progress in text-to-image (T2I) generative models (Ramesh et al., 2022; Saharia et al., 2022; Rombach et al., 2022b; Podell et al., 2023; Dai et al., 2023b). The photorealistic quality and aesthetics of generated images has positioned T2I generative models at the center of the current AI revolution. However, progress in image quality has come at the expense of model representation diversity and prompt-image consistency (Hall et al., 2023). From a user's perspective, the fact that not all the elements of the input text prompt are properly represented in the generated image is particularly problematic as it induces a tedious and time-consuming trial-and-error process with the T2I model to refine the initial prompt in order to generate the originally intended image. Common consistency failure modes of the T2I generative models include missing objects, wrong object cardinality, missing or mixed object attributes, and non-compliance with requested spatial relationships among objects in the image (Wu et al., 2023a).

To improve prompt-image consistency, researchers have explored avenues such as adjusting the sampling guidance scale (Ho & Salimans, 2022), modifying cross-attention operations (Feng et al., 2022; Epstein et al., 2023; Liu et al., 2022; Chefer et al., 2023; Wu et al., 2023a), fine-tuning models (Lee et al., 2023; Wu et al., 2023c; Sun et al., 2023), leveraging additional input modalities such as layouts (Cho et al., 2023b; Lian et al., 2023), and selecting images post-hoc (Karthik et al., 2023). Most of these approaches require access to the model's weights and are not applicable when models are only accessible through an API interface, *e.g.*, (Betker et al., 2023). The only two approaches that are applicable to API-accessible models are guidance scale modification and post-hoc image selection. However, these approaches rely on a single text prompt – the one provided by the user – so their ability to generate diverse images is limited to resampling of the input noise, *i.e.*, changing the random seed. Perhaps more importantly, both these approaches have unfavorable trade-offs, as they improve prompt-image consistency at the expense of image quality and diversity – *e.g.*, high guidance scales lead to reduced image quality and diversity, while post-hoc selecting the most consistent images decreases significantly the representation diversity.

At the same time, the natural language processing (NLP) community has explored large language models (LLMs) as prompt optimizers for NLP tasks (Pryzant et al., 2023; Yang et al., 2023; Guo et al., 2023; Fernando et al., 2023), showing that LLMs can relieve humans from the manual and tedious task of prompt-engineering. One particularly interesting approach is in-context learning (ICL) (Dong et al., 2023), where an LLM can learn to solve a new task from just a handful of in-context examples provided in the input prompt. Notably, LLMs can accomplish this without requiring parameter updates – *e.g.*, LLMs can solve simple regression tasks when provided with a few input-output examples (Mirchandani et al., 2023). ICL enables rapid task adaptation by changing the problem definition directly in the prompt. However, ICL commonly relies on predefined datasets of examples (input-output pairs), which might be challenging to obtain. To the best of our knowledge, ICL has not yet been explored in the context of T2I generative models, which pose unique challenges: (1) the in-context examples of prompts and the associated scores are not readily available, and (2) the LLM needs to ground the prompt in the generated images in order to understand how to refine it. Recently, and once again in the NLP domain, *optimization-by-prompting* (OPRO) (Yang et al., 2023) has extended previous ICL approaches by iteratively optimizing instruction prompts to maximize task accuracy based on a dataset of input-output examples, leveraging feedback from past instructions and their respective task accuracies.

In this work, we propose the first ICL-based method to improve prompt-image consistency in T2I models. In particular, we leverage optimization-by-prompting and introduce a novel inference-time optimization framework for T2I prompts, which constructs a dataset of in-context examples *on-the-fly*. Our framework, *OPtimization for T2I generative models* (OPT2I), involves a pre-trained T2I model, an LLM, and an automatic prompt-image consistency score – *e.g.*, CLIPScore (Hessel et al., 2021) or Davidsonian Scene Graph score (DSG) (Cho et al., 2023a). Through ICL, the LLM iteratively improves a user-provided text prompt by suggesting alternative prompts that lead to images that are more aligned with the user's intention, as measured by the consistency score. At each optimization iteration, the in-context examples are updated to include the best solutions found so far. Intuitively, we expect our method to explore the space of possible prompt paraphrases and gradually discover the patterns in its previously suggested prompts that lead to highly-consistent images. Crucially, the optimized prompts will produce more consistent images on expectation, across multiple input noise samples. OPT2I is designed to be a versatile approach that works as a *plug-and-play* solution with diverse T2I models, LLMs, and scoring functions since it does not require any parameter updates. An overview of our framework is presented in Figure 1.

Through extensive experiments, we show that OPT2I consistently outperforms paraphrasing baselines (*e.g.*, random paraphrasing and Promptist (Hao et al., 2022)), and boosts the prompt-image consistency by up to 12.2% and 24.9% on MSCOCO (Lin et al., 2014) and PartiPrompts (Yu et al., 2022) datasets, respectively. Notably, we achieve this improvement while preserving the Fréchet Inception Distance (FID) (Heusel et al., 2017) and increasing the recall between generated and real data. Moreover, we observe that OPT2I achieves consistent improvements for diverse T2I models and is robust to the choice of LLM. Our qualitative results reveal that the optimized prompts oftentimes emphasize the elements of the initial prompt that do not appear in the generated images by either providing additional details about those or reordering elements in the prompt to place the ignored elements at the beginning. In summary, our contributions are:

- We propose OPT2I, a training-free T2I optimization-by-prompting framework that provides refined prompts for a T2I model that improve prompt-image consistency.

- OPT2I is a versatile framework as it is not tied to any particular T2I model and is robust to both the choice of LLM as well as consistency metric.

- We show that OPT2I consistently outperforms paraphrasing baselines and can boost the prompt-image consistency by up to 24.9%.

## 2 OPT2I: Optimization by prompting for T2I

Figure 2 depicts our T2I *optimization-by-prompting* framework, which is composed of a pre-trained T2I model and an LLM that work together to optimize a prompt-image *consistency score*. Our framework starts from a *user prompt* and iteratively generates *revised prompts* with the goal of maximizing the chosen

consistency score. More concretely, we start by feeding the user prompt to the T2I model to generate multiple images. Next, we compute the consistency score between each generated image and the user prompt, and average the scores. We then initialize the *meta-prompt history* with the user prompt and its associated consistency score. Finally, we feed the meta-prompt to the LLM, which proposes a set of revised prompts. To start a new optimization step, we feed the revised prompts back to the T2I model that generate new images. Note that the consistency score is always computed w.r.t. the user prompt. At each optimization step, the meta-prompt history is updated to include the top-$k$ most consistent prompts (among revised prompts and user prompt), along with their consistency score. The prompt optimization process terminates when a maximum number of iterations is reached, or when a perfect/target consistency score is achieved. Finally, the best prompt is selected, namely, the *optimized prompt.*

## 2.1 Problem formulation

We assume access to an LLM, $f$, and a pre-trained T2I generative model, $g$. Given a text prompt, $p$, we can generate an image $I$ conditioned on the prompt with our T2I generator, $I = g(p)$. Let us define the set of all possible paraphrases from $p$ that can be obtained with an LLM as $\mathbb{P} = \{p_i\}$, and let us introduce a prompt-image consistency score, $\mathcal{S}(p, I)$. Our objective is to find a prompt paraphrase $\hat{p} \in \mathbb{P}$ that maximizes the expected consistency score of sampled images:

$$\hat{p} = \underset{p_i \sim \mathbb{P}}{\arg\max} \; \underset{I \sim g(p_i)}{\mathbb{E}} \left[ \mathcal{S}(p_0, I) \right],$$

where $p_0$ denotes the user prompt. We approach the optimization problem with ICL by iteratively searching for revised prompts generated by the LLM,

$$P_t = f(C(\{p_0\} \cup P_1, \ldots, \cup P_{t-1})),$$

where $P_i$ is the set of prompts generated at iteration $i$, $t$ is the current iteration, and $C$ is a function that defines the context of prompt-score pairs fed to the LLM.

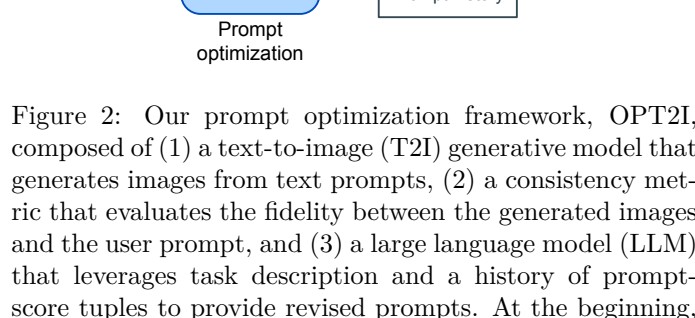

Figure 2: Our prompt optimization framework, OPT2I, composed of (1) a text-to-image (T2I) generative model that generates images from text prompts, (2) a consistency metric that evaluates the fidelity between the generated images and the user prompt, and (3) a large language model (LLM) that leverages task description and a history of prompt-score tuples to provide revised prompts. At the beginning, the revised prompt is initialized with the user prompt.

## 2.2 Meta-prompt design

We adapt LLMs for T2I prompt optimization via ICL. We denote *meta-prompt* the prompt which instructs the LLM to optimize prompts for T2I models. Our meta-prompt is composed of a *task instruction* and a *history* of past revised prompt-score pairs. The list of meta-prompts used can be found in Appendix A.

The meta-prompt provides context about T2I models, the consistency metric and the optimization problem. Additionally, it contains a history of prompt-score pairs that provides context about which paraphrases worked best in the past for a particular T2I model, encouraging the LLM to build upon the most successful prompts and removing the need for explicitly specifying how to modify the T2I prompt. The consistency score is normalized to an integer between 0 and 100, and we only keep in the history the top-$k$ scoring prompts found so far, sorted by increasing score.

### 2.3 Optimization objective

A critical part of our framework is feeding visual feedback to the LLM. The visual feedback is captured by the consistency score, which determines how good the candidate prompt is at generating consistent images. Although OPT2I can work with any – even non-differentiable – consistency score, we argue that the consistency score must be detailed enough for the LLM to infer how to improve the candidate prompt. While CLIPScore (Hessel et al., 2021) is arguably the most popular metric for measuring prompt-image consistency, in our initial experiments we found that scoring a prompt with a single scalar is too coarse for our purposes. Thus, we opt for two metrics that provide finer-grained information about the prompt-image consistency: (1) Davidsonian Scene Graph (DSG) score (Cho et al., 2023a), and (2) our proposed *decomposed CLIPScore*.

**DSG** assesses prompt-image consistency based on a question generation and answering approach, similar to TIFA (Hu et al., 2023). In particular, DSG generates atomic and unique binary questions from the user prompt that are organized into semantic dependency graphs. For example, "a bike lying on the ground, covered in snow" is decomposed into: (1) "Is there a bike?"; (2) "Is the bike lying on the ground?"; (3) "Is the bike covered in snow?". In this case, questions (2) and (3) depend on (1), and so (1) is used to validate (2) and (3). These questions are then answered by an off-the-shelf VQA model based on the generated image. We include the resulting question-answer pairs in our meta-prompt. A global score per prompt-image pair is computed by averaging across answer scores.

**Decomposed CLIPScore** computes a partial consistency score for each noun phrase present in the user prompt. For example, "a bike lying on the ground, covered in snow" is decomposed into "a bike", "the ground" and "snow". Each noun phrase is then scored against the generated image using CLIPScore, resulting in a list of pairs of noun phrases and their associated scores, which are included in our meta-prompt. A global score per prompt-image pair is computed by averaging across subscores. We provide examples of decomposed CLIPScore and DSG outputs in Appendix A.

### 2.4 Exploration-exploitation trade-off

During the optimization process, OPT2I requires controllability over the LLM's exploration-exploitation trade-off, as the LLM could either focus on exploring possible revised prompts or on exploiting the context provided in the meta-prompt history. On the one hand, too much exploration would hamper the optimization as it could be hard to find a high quality solution. On the other hand, too much exploitation would limit the exploration to prompts that are very similar to the ones already presented in the meta-prompt history. We control this balance by adjusting the number of generated revised prompts per iteration and the LLM sampling temperature. Moreover, as our objective is to find prompts that work well across different T2I input noise samples, we generate multiple images per prompt at each iteration.

## 3 Experiments

We first introduce our experimental setting. Next, we validate the effectiveness of OPT2I in improving prompt-image consistency, compare it to paraphrasing baselines (random paraphrasing and Promptist), and show some qualitative results. Then, we explore the trade-offs with image quality and diversity. And finally, we ablate OPT2I components and touch on post-hoc image selection.

### 3.1 Experimental setting

**Benchmarks.** We run experiments using prompts from MSCOCO (Lin et al., 2014) and PartiPrompts (P2) (Yu et al., 2022). For MSCOCO, we use the 2000 captions from the validation set as in (Hu et al., 2023). These captions represent real world scenes containing common objects. PartiPrompts, instead, is a collection of 1600 artificial prompts, often unrealistic, divided into categories to stress different capabilities of T2I generative models. We select our PartiPrompts subset by merging the first 50 prompts from the most challenging categories: "Properties & Positioning", "Quantity", "Fine-grained Detail", and "Complex". This results in a set of 185 complex prompts.

**Baselines.** We compare OPT2I against a random paraphrasing baseline, where the LLM is asked to generate diverse paraphrases of the user prompt, without any context about the consistency of the images generated from it. The meta-prompt used to obtain paraphrases is provided in Appendix A. We also compare to Promptist Hao et al. (2022), which relies on a dataset of initial and target prompts to finetune an LLM to rewrite user prompts with the goal of improving image *aesthetics*, while trying to maintain prompt-image consistency.

**Evaluation metrics.** We measure the quality of a T2I prompt by averaging prompt-image consistency scores across multiple image generations (*i.e.*, multiple random seeds for the initial noise). For each generated image, we compute its consistency with the user prompt. We consider our proposed *decomposed CLIPScore* (dCS) and the recent DSG score (Cho et al., 2023a) as consistency metrics (see Section 2.3 for more details). For DSG score, we use Instruct-BLIP (Dai et al., 2023a) as the VQA model. To assess the trade-off between prompt-image consistency, image quality and diversity, we additionally compute FID score (Heusel et al., 2017), precision and recall metrics (Naeem et al., 2020).

**LLMs and T2I models.** For the T2I model, we consider (1) a state-of-the-art latent diffusion model, namely `LDM-2.1` (Rombach et al., 2022a), which uses a CLIP text encoder for conditioning, and (2) a cascaded pixel-based diffusion model, `CDM-M`, which instead relies on the conditioning from a large language model, `T5-XXL` (Raffel et al., 2020), similarly to (Saharia et al., 2022). For the LLM, we experiment with the open source `Llama-2-70B- chat` (`Llama-2`) (Touvron et al., 2023) and with `GPT-3.5-Turbo-0613` (`GPT-3.5`) (Brown et al., 2020).

**Implementation details.** Unless stated otherwise, OPT2I runs for at most 30 iterations generating 5 new revised prompts per iteration, while the random paraphrasing baseline generates 150 prompts at once (see Section 3.4 for a discussion about the relationship between #iters and #prompts/iter). We instruct the LLM to generate revised prompts by enumerating them in a single response to prevent duplication, rather than making multiple calls with a sampling temperature greater than 0. In the optimization meta-prompt, we set the history length to 5. To speed up image generation, we use DDIM (Song et al., 2020) sampling. We perform 50 inference steps with `LDM-2.1`, while for `CDM-M` we perform 100 steps with the low-resolution generator and 50 steps with the super-resolution network – following the default parameters in both cases. The guidance scale for both T2I models is kept to the suggested value of 7.5. Finally, in our experiments, we fix the initial random seeds across iterations and prompts wherever possible, *i.e.* we fix 4 random seeds for sampling different prior/noise vectors to generate 4 images from the same prompt. However, we note that `CDM-M` does not allow batched generation with fixed seeds. Experiments without seed-fixing are reported in Appendix B as we observe no substantial differences.

## 3.2 Main results

**T2I optimization by prompting.** In Figure 3, we plot the prompt optimization curves with `LDM-2.1/CDM-M` as T2I models, `Llama-2/GPT-3.5` as LLM, and decomposed CLIPscore (dCS)/DSG as the scorer for prompts from MSCOCO and PartiPrompts. Each data point corresponds to the *mean/max* relative improvement in consistency score (w.r.t. the user prompt) achieved by revised prompts generated in that iteration, averaged across the full dataset of prompts. The optimization curves show an overall upward trend, which confirms that the LLM in OPT2I is capable of optimizing T2I prompts. These improvements are especially noticeable in the *max* consistency score. The initial dip in *mean* consistency score is expected due to the initial exploration, since the LLM has limited context provided only by the user prompt (1-shot ICL). As the optimization progresses, the LLM generates more consistent revised prompts at each iteration, as its context is increasingly enriched with the performance of previous revised prompts. Notably, achieving a positive *mean* relative consistency starting from a single in-context example is a very challenging task (Wei et al., 2023), and OPT2I achieves this goal for all configurations except for `Llama-2` and `GPT-3.5` in the *dCS(P2)* setting (Figure 3b), where it only get close to 0. As mentioned Section 3.1, PartiPrompts is a hard benchmark containing highly detailed and complex prompts, so it is perhaps unsurprising that decomposed CLIPScore falls short (with improvements $< 7\%$ in the max case, and $< 2\%$ in the mean case) – given the imperfect decomposition into noun-phrases. We also explored a hierarchical version of decomposed CLIPscore leveraging constituency trees, which did not show any improvement over our noun-phrase based decomposition, further reinforcing the criticism that CLIP behaves as a bag-of-words and is unable to properly capture object attributes and relations (Yuksekgonul et al., 2022; Yamada et al., 2022). Instead, using

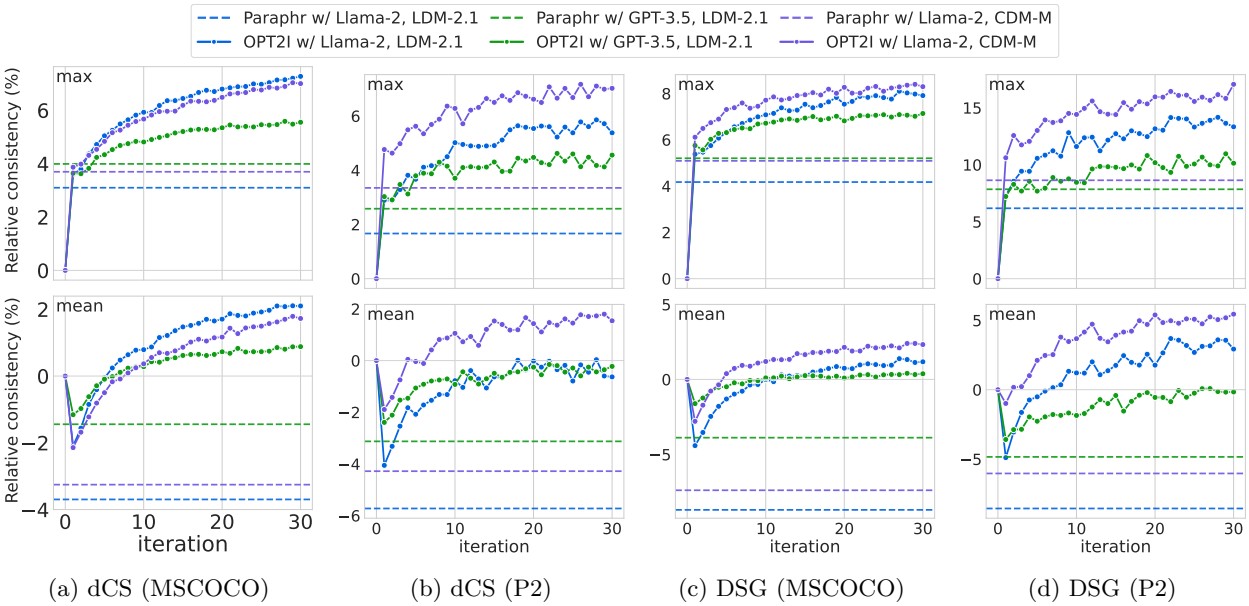

Figure 3: OPT2I curves with different consistency objectives (dCS vs. DSG), LLMs, and T2I models. Each plot track either the *max* or the *mean* relative improvement in consistency across revised prompts per iteration.

a more detailed consistency score during the prompt optimization process, such as DSG, results in more significant improvements ($< 17\%$ in the max case, and $< 5\%$ in the mean case).

**Comparison to paraphrasing baselines.** Table 1 shows our proposed OPT2I framework is robust to the choice of LLM, T2I model and optimization/evaluation objective. In particular, for dCS, we report relative improvement as $score_{best}/score_{init} - 1$. For DSG score, since the initial score can be zero and it is already a percentage, we instead report $score_{best} - score_{init}$. We observe that OPT2I consistently outperforms the random paraphrasing baseline across different LLMs and T2I models. Additionally, we can see that both paraphrasing and optimization get around a 10% boost in consistency improvement when using DSG as optimization objective instead of dCS for P2 but not for MSCOCO. This highlights again that more complex prompts, such as those from PartiPrompts, benefit from a more accurate consistency metric. We note that some prompts might already have a fairly high initial consistency score (see App. B.2), so there is little room for improvement. For instance, prompts from MSCOCO evaluated with DSG have an average initial score of 86.54%, which means that the improvement in this setting has an upper bound of 13.46%.

In addition to random paraphrasing, we compare OPT2I to Promptist (Hao et al., 2022) on MSCOCO prompts by generating images from initial/best prompts (4 images/prompt) with `SD-1.4` (Promptist's reference model) and `LDM-2.1`, evaluating consistency with DSG score. We observe Promptist decreases the consistency score by $-3.56\%/-3.29\%$ on `SD-1.4`/`LDM-2.1`, while OPT2I (`Llama-2`) improves consistency by $+14.16\%/+11.21\%$. This aligns with the results reported in (Hao et al., 2022), which show that optimizing prompts primarily for aesthetics actually decreases prompt-image consistency.

**Qualitative results.** In Figure 4, we provide examples of images generated from user and optimized prompts with OPT2I for different LLMs and T2I models. We observe OPT2I is capable of finding paraphrases of the user prompt which considerably improve the consistency between the generated images and the initial, user-provided prompt, as measured by DSG in this case. These examples suggest the optimized prompts are capable of steering the T2I model towards generating visual elements that were ignored with the initial phrasing. From our qualitative analysis, we observed the LLM uses several strategies to emphasize the missing visual elements, such as providing a more detailed description of those elements (*e.g.*, "a flower" $\rightarrow$ "a vibrant flower arrangement", "a vase filled with fresh blossoms") or placing them at the beginning of the sentence (*e.g.*, "four teacups surrounding a kettle" $\rightarrow$ "surround a kettle placed at the center with four

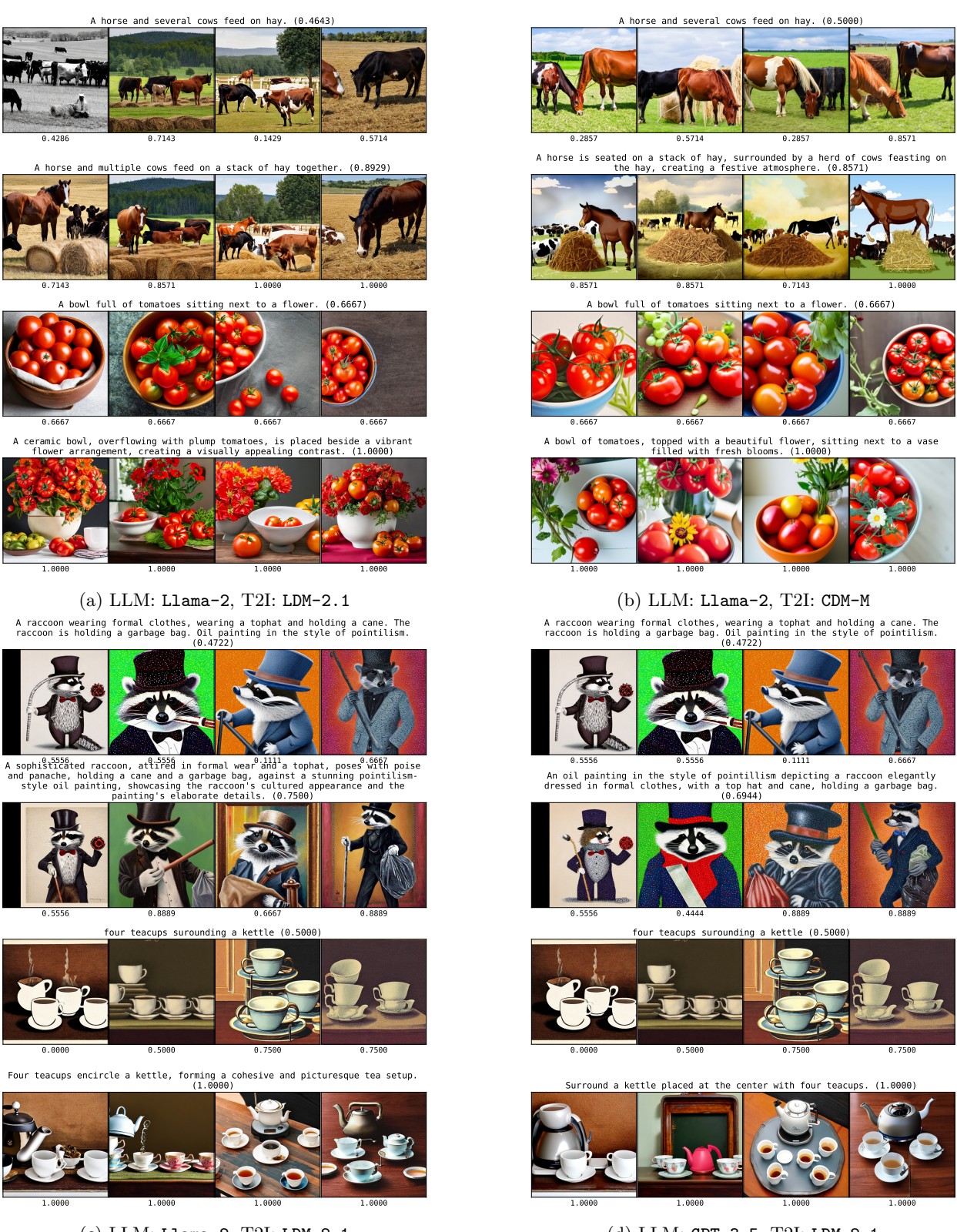

Figure 4: Selected qualitative results for prompts from MSCOCO (a-b) and P2 (c-d) datasets, using DSG as consistency metric. For each setup, we display four rows (from the top): initial prompt #1, optimized prompt #1, initial prompt #2, and optimized prompt #2. Each column corresponds to a different T2I model random seed. We report average consistency score across seeds in between parenthesis.

Table 1: Relative improvement in T2I consistency between the user prompt and the best prompt found, averaged across all prompts in the dataset. Every method generates the same total number of prompts (and images).

| Method | Objective | LLM | T2I | Dataset | |
|---|---|---|---|---|---|
| | | | | MSCOCO | P2 |
| Paraphrasing OPT2I | dCS | Llama-2 | LDM-2.1 | +9.96 +11.88 | +8.00 +10.34 |
| Paraphrasing OPT2I | DSG | Llama-2 | LDM-2.1 | +10.67 +11.21 | +19.22 +22.24 |
| Paraphrasing OPT2I | dCS | GPT-3.5 | LDM-2.1 | +9.81 +10.35 | +8.06 +9.33 |
| Paraphrasing OPT2I | DSG | GPT-3.5 | LDM-2.1 | +10.53 +10.85 | +19.09 +19.77 |
| Paraphrasing OPT2I | dCS | Llama-2 | CDM-M | +11.00 +12.21 | +10.29 +12.13 |
| Paraphrasing OPT2I | DSG | Llama-2 | CDM-M | +10.53 +11.07 | +22.24 +24.93 |

Table 2: Distributional metrics on the MSCOCO dataset.

| Prompt | Obj. | LLM | T2I | FID (↓) | | Prec. (↑) | | Rec. (↑) | |
|---|---|---|---|---|---|---|---|---|---|
| | | | | IV3 | CLIP | IV3 | CLIP | IV3 | CLIP |
| Initial | - | - | LDM-2.1 | 34.6 | 16.0 | **81.6** | **85.9** | 47.2 | 22.8 |
| OPT2I | dCS | Llama-2 | LDM-2.1 | 34.6 | 14.9 | 70.7 | 81.0 | **55.6** | **30.0** |
| OPT2I | DSG | Llama-2 | LDM-2.1 | **33.3** | 15.4 | 79.0 | 84.3 | 54.4 | 27.0 |
| OPT2I | dCS | GPT-3.5 | LDM-2.1 | 34.1 | **14.3** | 74.9 | 84.0 | 53.9 | 27.3 |
| OPT2I | DSG | GPT-3.5 | LDM-2.1 | **33.4** | 15.6 | 80.3 | 85.4 | 50.5 | 21.7 |
| Initial | - | - | CDM-M | 41.2 | **15.2** | **82.2** | **85.6** | 38.8 | 26.0 |
| OPT2I | dCS | Llama-2 | CDM-M | **39.8** | 15.2 | 77.1 | 80.9 | **45.4** | **29.5** |
| OPT2I | DSG | Llama-2 | CDM-M | 39.9 | 15.2 | 79.6 | 82.5 | 39.9 | 25.0 |

Table 3: Meta-prompt ablations.

| Conciseness | Prioritize | Reasoning | Structure | dCS (%) |
|---|---|---|---|---|
| ✓ | ✗ | ✗ | ✗ | +9.68 |
| ✓ | ✓ | ✗ | ✗ | **+10.34** |
| ✓ | ✓ | ✓ | ✗ | +10.23 |
| ✓ | ✓ | ✗ | ✓ | +9.99 |

teacups"). We note a perfect consistency score does not ensure perfectly aligned images (*e.g.*, for the user prompt "four teacups surrounding a kettle", all optimized prompts reach a DSG score of 100% while the cardinality of teacups remains incorrect), which highlights the limitations of current prompt-image consistency scores. We also observe that prompts optimized by Llama-2 tend to be longer than those from GPT-3.5 (see App. B.5), and that images generated by CDM-M from user prompts are generally more consistent than those generated by LDM-2.1, which we attribute to the use of a stronger text encoder (T5-XXL instead of CLIP).

## 3.3 Trade-offs with image quality and diversity

Following common practice in the T2I community, we evaluate the quality of OPT2I generations by computing image generation metrics such as FID, precision (P), and recall (R). We use the 2000 prompts from the MSCOCO validation set that are included in the TIFAv1 benchmark (Hu et al., 2023), and generate 4 images for each initial and best prompt. To ensure robust conclusions, we use two feature extractors in our metrics: Inception-v3 (IV3) (Szegedy et al., 2016) and CLIP (Radford et al., 2021). Results in Table 2 show that the FID of prompts optimized with OPT2I is either on-par or better compared to that of initial prompts, validating that our method does not trade image quality for consistency. Hence, we conclude FID is not affected by our optimization strategy. However, in terms of precision and recall, we observe that optimized prompts reach higher recall at the expense of lower precision compared to the user prompt. This can be explained by the fact that rephrasing the input prompt allows to generate more diverse images (higher recall), which may occasionally fall outside of the manifold of natural images (lower precision); *e.g.*, in Fig. 12 (Appendix B), optimizing for consistency leads to a change of artistic style. The observed trade-off between realism and diversity aligns with the results reported by Astolfi et al. (2024).

## 3.4 Ablations

We perform ablations with Llama-2 and LDM-2.1 on PartiPrompts using default parameters unless otherwise specified. Figure 5 illustrates the trade-off between exploration and exploitation, implemented as the number of revised prompts per iteration (#prompts/iter) and the number of optimization iterations (#iterations), respectively. Generating more prompts at each iteration increases the *exploration* of multiple solutions given the same context, while by increasing the number of iterations, the LLM can *exploit* more frequent feedback from the T2I model and the consistency score. We observe that increasing number of iterations leads to a higher consistency improvement. In other words, more exploitation is beneficial with a fixed budget of 150 prompt generations. However, pushing exploitation too much, *i.e.*, #it = 150 and #p/it = 1, becomes harmful.

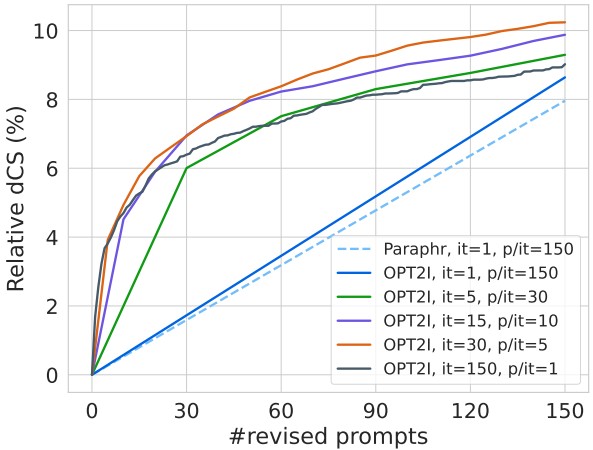
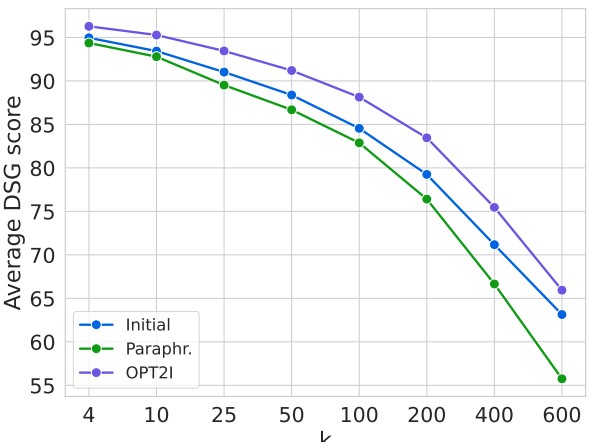

Figure 5: Cumulative *max* relative dCS as a function of #revised prompts = #iterations · #prompts/iter.

Figure 6: Average DSG score for the top-$k$ most consistent images among 600.

In table 3, we report relative consistency (dCS) when ablating for different task instructions in the meta-prompt. We explore four instruction additions to be combined with our base meta-prompt. *Conciseness* encourages to explore paraphrases beyond just adding details/adjectives; *Prioritize* encourages focusing on missing/low-score elements; *Reasoning* encourages to reason about the in-context examples; and *Structure* asks for simple vocabulary informative of the structure of images, *e.g.*, foreground and background (full meta-prompts are provided in Appendix A). We observe that *Prioritize* achieves slightly better performance over *Reasoning* and *Structure*, yet the LLM remains fairly robust to specific meta-prompt phrasing.

### 3.5  Post-hoc image selection

We emphasize OPT2I aims to optimize *prompts* to generate more consistent *images on expectation*. However, for the sake of completeness, we also evaluate the setting where we generate the same amount of images from the initial prompt and select the most consistent ones. In particular, we generate 600 images from PartiPrompts using either random image sampling from the initial prompt, paraphrasing, or OPT2I, and select the top-$k$ most consistent images based on DSG score. In Fig. 6, we observe that OPT2I consistently outperforms both baselines. Interestingly, sampling from the initial prompt outperforms paraphrasing, which might be due to random paraphrases deviating too much from the user's intent.

## 4   Related work

**Improving consistency in T2I models.** Several recent works propose extensions to T2I models to improve their faithfulness to user prompts. Some studies focus on improving the guidance with cross-attention (Feng et al., 2022; Epstein et al., 2023; Liu et al., 2022; Chefer et al., 2023; Wu et al., 2023a). Other studies first convert a textual prompt into a layout before feeding it to a layout-to-image generative model (Cho et al., 2023b; Lian et al., 2023). Recent works also finetune T2I models on human (Lee et al., 2023; Wu et al., 2023c; Wallace et al., 2023) or AI model (Sun et al., 2023) feedback, or perform post-hoc image selection (Karthik et al., 2023). In contrast, OPT2I acts exclusively at the level of input prompt in text space, without accessing model weights, making it applicable to a wider range of T2I models, including those only accessible through an API.

**LLMs as prompt optimizers.** Several recent works explore the role of LLMs as prompt optimizers for NLP tasks. Some use LLMs to directly optimize the task instruction for ICL (Zhou et al., 2022; Pryzant et al., 2023; Yang et al., 2023). Other studies use LLMs to mutate prompts for evolutionary algorithms (Guo et al., 2023; Fernando et al., 2023). A crucial difference between these works and our method is that they optimize a task instruction prompt by using a training set, which is subsequently applied across test examples, while we

perform multimodal inference-time optimization on individual T2I prompts. More similar to our work, other studies rewrite prompts for T2I models using an LLM. (Hao et al., 2022) finetunes an LLM with reinforcement learning to improve image aesthetics, while (Valerio et al., 2023) focuses on filtering out non-visual prompt elements. In contrast, OPT2I aims to improve prompt-image consistency via optimization-by-prompting.

**Evaluating prompt-image consistency.** Several metrics have been proposed to evaluate prompt-image consistency. CLIPScore (Hessel et al., 2021) is the *de facto* standard for measuring the compatibility of image-caption pairs, used both for image captioning and text-conditioned image generation. However, CLIPScore provides a single global score, which can be too coarse to understand failures in the generated images. Consequently, subsequent metrics such as TIFA (Hu et al., 2023), VQ$^2$ (Yarom et al., 2023) or DSG (Cho et al., 2023a) propose generating pairs of questions and answers from T2I prompts and using off-the-shelf VQA models to evaluate each of them on the generated images, providing a fine-grained score. Other recent studies suggest directly learning a prompt-image consistency metric from human feedback (Xu et al., 2023; Wu et al., 2023b; Kirstain et al., 2023). However, none of these metrics are without flaws and human judgment remains the most reliable way of evaluating prompt-image consistency.

## 5  Conclusions

In this paper, we introduced the first T2I optimization-by-prompting framework to improve prompt-image consistency. Through extensive evaluations, we showed that OPT2I can be effectively applied to different combinations of LLM, T2I models and consistency metrics, consistently outperforming paraphrasing baselines and yielding prompt-image consistency improvements of up to 24.9% over the user prompt, while maintaining the FID between generated and real images. By contrasting MSCOCO and PartiPrompts results, we highlighted the importance of the choice of consistency score: complex prompts in PartiPrompts appear to significantly benefit from more detailed scores such as DSG. Qualitatively, we observed that optimizing prompts for prompt-image consistency oftentimes translates into emphasizing initially ignored elements in the generated images, by either providing additional details about those or rewording the prompt such that the ignored elements appear at the beginning. Interestingly, such prompt modifications steer the generated images away from the learned modes, resulting in a higher recall w.r.t. the real data distribution.

**Limitations.** One limitation of our method is that it expects prompt-image consistency scores to work reasonably well. However, this assumption might not hold in some cases. For instance, it has been shown that CLIP (used for CLIPScore) sometimes behaves like a bag-of-words (Yuksekgonul et al., 2022; Yamada et al., 2022). VQA-based prompt-image consistency metrics such as TIFA or DSG also suffer from limitations in generating questions (*e.g.*, the question "Is the horse on the hay?" is generated from the prompt "A horse and several cows feed on hay.") or in answering them with a VQA model (*e.g.*, for the prompt "A bowl full of tomatoes sitting next to a flower.", the VQA model answers that there is a flower when it is in fact a bouquet made of tomatoes). Moreover, using these metrics as optimization objectives might exacerbate their failure modes by finding prompts which generate images that fulfill the requirements for a high score in an adversarial way. This highlights the need for further research in developing more robust prompt-image consistency metrics which can be used as optimization objectives in addition to evaluation.

Another limitation of our approach is its runtime, which is a consequence of performing inference-time optimization. For instance, running the optimization process with `Llama-2`, `LDM-2.1` and DSG score, generating 5 prompt paraphrases per iteration and 4 images per prompt with 50 diffusion steps, takes 7.34/20.27 iterations on average for COCO/PartiPrompts, which translates to ∼10/28 minutes when using NVIDIA V100 GPUs. However, we emphasize that (1) OPT2I is designed to be a versatile approach that works as a *plug-and-play* solution with diverse T2I models and LLMs since it does not require any parameter updates nor training data, and (2) optimizing T2I prompts with our automatic framework relieves humans from the manual and tedious task of prompt-engineering.

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

# A  Additional method details

## A.1  Meta-prompts

Tables 4-7 include all the prompts used in our work. The teal text is fed to the LLM as a system prompt, and the purple text denotes placeholders to be dynamically filled.

Table 4: Meta-prompt used for decomposing prompts into noun phrases for dCS.

```
Decompose the following sentence into individual noun phrases. Ignore prefixes such as 'a photo of', 'a
picture of', 'a portrait of', etc. Your response should only be a list of comma separated values, eg: '
foo, bar, baz'

{prompt}
```

Table 5: Meta-prompt used for the paraphrasing baseline.

```
Generate {num_solutions} paraphrases of the following image description while keeping the semantic
meaning: "{user_prompt}". Respond with each new prompt in between <PROMPT> and </PROMPT>, eg:

1. <PROMPT>paraphrase 1</PROMPT>
2. <PROMPT>paraphase 2</PROMPT>
...
{num_solutions}. <PROMPT>paraphrase {num_solutions}</PROMPT>
```

Table 6: Meta-prompt used for OPT2I with dCS (top) or DSG (bottom) as scorer.

```
You are an expert prompt optimizer for text-to-image models. Text-to-image models take a text prompt as
input and generate images depicting the prompt as output. You translate prompts written by humans into
better prompts for the text-to-image models. Your answers should be concise and effective.

Your task is to optimize this initial prompt written by a human: "{user_prompt}". Below are some previous
 prompts with a decomposition of their visual elements. Each element is paired with a score indicating
its presence in the generated image. The prompts are arranged in ascending order based on their scores,
which range from 0 to 100. Higher scores indicate higher likelihood of presence.

1. {revised_prompt_1}
score: {avg_score_1}
visual elements:
{subprompt_1_1} {clip_score_1_1}
{subprompt_1_2} {clip_score_1_2}
(... more questions ...)

(... more examples ...)

Generate {num_solutions} paraphrases of the initial prompt which keep the semantic meaning and that have
higher scores than all the prompts above. Prioritize optimizing for object with lowest scores. Favor
substitutions and reorderings over additions. Respond with each new prompt in between <PROMPT> and </
PROMPT>, eg:

1. <PROMPT>paraphrase 1</PROMPT>
2. <PROMPT>paraphase 2</PROMPT>
...
{num_solutions}. <PROMPT>paraphrase {num_solutions}</PROMPT>
```

```
You are an expert prompt optimizer for text-to-image models. Text-to-image models take a text prompt as
input and generate images depicting the prompt as output. You translate prompts written by humans into
better prompts for the text-to-image models. Your answers should be concise and effective.

Your task is to optimize this initial prompt written by a human: "{user_prompt}". Below are some previous
 prompts with the consistency of each prompt's visual elements in the generated image via a set of binary
 questions. The prompts are arranged in ascending order based on their overall consistency score, which
ranges from 0 to 100 (higher is better).

1. {revised_prompt_1}
overall score: {dsg_score_1}
evaluation questions:
{question_1_1} {vqa_score_1_1}
{question_1_2} {vqa_score_1_2}
(... more questions ...)

(... more examples ...)

Generate {num_solutions} paraphrases of the initial prompt which keep the semantic meaning and that have
higher scores than all the prompts above. Focus on optimizing for the visual elements that are not
consistent. Favor substitutions and reorderings over additions. Respond with each new prompt in between <
PROMPT> and </PROMPT>, eg:

1. <PROMPT>paraphrase 1</PROMPT
2. <PROMPT>paraphase 2</PROMPT>
...
{num_solutions}. <PROMPT>paraphrase {num_solutions}</PROMPT>
```

Table 7: Meta-prompt ablations. Modifications w.r.t. the base meta-prompt are denoted in blue.

Conciseness:

```
(... more examples ...)

Generate {num_solutions} paraphrases of the initial prompt which keep the semantic meaning and that have
higher scores than all the prompts above. Favor substitutions and reorderings over additions. Respond
with each new prompt in between <PROMPT> and </PROMPT>, eg:
...
```

Conciseness + prioritize:

```
(... more examples ...)

Generate {num_solutions} paraphrases of the initial prompt which keep the semantic meaning and that have
higher scores than all the prompts above. Prioritize optimizing for object with lowest scores. Favor
substitutions and reorderings over additions. Respond with each new prompt in between <PROMPT> and </
PROMPT>, eg:
...
```

Conciseness + prioritize + reasoning:

```
(... more examples ...)

Briefly reason (max two sentences) about the prompts above to understand why certain objects have higher
or lower scores in certain prompts. Then, based on this reasoning, generate {num_solutions} paraphrases
of the initial prompt which keep the semantic meaning and that have higher scores than all the prompts
above. Prioritize optimizing for objects with lowest scores while keeping high scores for the other
objects. Favor substitutions and reorderings over additions. Respond with each new prompt in between <
PROMPT> and </PROMPT>, eg:
...
```

Conciseness + prioritize + structure:

```
(... more examples ...)

Generate {num_solutions} paraphrases of the initial prompt which keep the semantic meaning and that have
higher scores than all the prompts above. PRIORITIZE optimizing for objects with lowest scores while
keeping high scores for the other objects. FAVOR substitutions and reorderings over additions. USE simple
 words/concepts, understable from a text-to-image model, e.g., distinguish foreground and background.
Respond with each new prompt in between <PROMPT> and </PROMPT>, eg:
...
```

### A.2 Examples of prompt decompositions

Table 8 shows a few examples of outputs generated by dCS (noun phrases) and DSG (binary questions) from the input prompt. These outputs are then used to compute a fine-grained consistency score w.r.t. the generated image, either with a multimodal encoder (dCS) or a VQA model (DSG).

Table 8: Prompt decompositions into noun phrases (dCS) and binary questions (DGS).

| Prompt | dCS | DSG |
|---|---|---|
| "A ginger cat is sleeping next to the window." | "ginger cat", "window" | "Is there a cat?", "Is the cat ginger?", "Is the cat sleeping?", "Is there a window?", "Is the cat next to the window?" |
| "Many electronic wires pass over the road with few cars on it." | "electronic wires", "road", "cars" | "Are there many electronic wires?", "Is there a road?", "Are there few cars?", "Do the electronic wires pass over the road?", "Are the cars on the road?" |
| "there is a long hot dog that has toppings on it" | "long hot dog", "toppings" | "Is there a hot dog?", "Is the hot dog long?", "Is the hot dog hot?", "Are there toppings on the hot dog?" |
| "the mona lisa wearing a cowboy hat and screaming a punk song into a microphone" | "the mona lisa", "a cowboy hat", "a punk song", "a microphone" | "Is there the Mona Lisa?", "Is the Mona Lisa wearing a cowboy hat?", "Is the Mona Lisa holding a microphone?", "Is the hat a cowboy hat?", "Is the Mona Lisa screaming?", "Is the song the Mona Lisa is singing punk?", "Is the Mona Lisa screaming into the microphone?" |
| "a photograph of a bird wearing headphones and speaking into a microphone in a recording studio" | "photograph", "bird", "headphones", "microphone", "recording studio" | "Is this a photograph?", "Is there a bird?", "Does the bird have headphones?", "Does the bird have a microphone?", "Is there a recording studio?", "Is the bird speaking into the microphone?", "Is the bird wearing headphones?", "Is the bird in the recording studio?" |
| "concentric squares fading from yellow on the outside to deep orange on the inside" | "concentric squares", "yellow", "outside", "deep orange", "inside" | "Are there squares?", "Are the squares concentric?", "Is the outer square yellow?", "Is the inner square deep orange?", "Is the inner square inside the outer square?", "Is the outer square on the outside?", "Is the inner square on the inside?" |

# B    Additional results

Table 9: Comparison with 1-shot in-context learning (ICL). We report relative improvement (%) in prompt-image consistency between the user prompt and the best prompt found, averaged across all prompts in the dataset.

| Method | Objective | LLM | T2I | Dataset | |
|---|---|---|---|---|---|
| | | | | MSCOCO | P2 |
| Paraphrasing | dCS | Llama-2 | LDM-2.1 | +9.86 | +8.00 |
| 1-shot ICL | | | | +10.67 | +8.74 |
| OPT2I | | | | **+11.68** | **+10.34** |
| Paraphrasing | DSG | Llama-2 | LDM-2.1 | +9.92 | +19.22 |
| 1-shot ICL | | | | +10.14 | +19.69 |
| OPT2I | | | | **+11.02** | **+22.24** |
| Paraphrasing | dCS | GPT-3.5 | LDM-2.1 | +9.64 | +8.06 |
| 1-shot ICL | | | | +9.21* | +8.72 |
| OPT2I | | | | **+10.56** | **+9.33** |
| Paraphrasing | DSG | GPT-3.5 | LDM-2.1 | +10.21 | +19.09 |
| 1-shot ICL | | | | +10.09* | +18.94* |
| OPT2I | | | | **+11.19** | **+19.77** |
| Paraphrasing | dCS | Llama-2 | CDM-M | +11.38 | +10.29 |
| 1-shot ICL | | | | +12.19 | +11.34 |
| OPT2I | | | | **+12.65** | **+12.13** |
| Paraphrasing | DSG | Llama-2 | CDM-M | +9.86 | +22.24 |
| 1-shot ICL | | | | +10.10 | +22.25 |
| OPT2I | | | | **+10.15** | **+24.93** |

## B.1    1-shot in-context learning as baseline

In this experiment, we compare OPT2I with 1-shot in-context learning (ICL), which we implement by running OPT2I with #iter = 1 and #prompts/iter = 150. Note that, in this setting, the LLM only receives feedback about the performance of the user prompt. We maintain the same experimental setting described in Section 3, except we use 200 prompts for MSCOCO, and report the results in Table 9. First, we notice that 1-shot ICL achieves higher prompt-image consistency than random paraphrasing in all settings except when using `GPT-3.5`, which performs on-par or slightly worse (marked with * in Table 9, see also the discussion in Section B.5). Second, and more importantly, we observe that OPT2I outperforms the 1-shot ICL baseline regardless of the consistency objective, LLM, or T2I model adopted. These results reinforce our previous claims: (1) the iterative procedure allows OPT2I to keep improving revised prompts over time, and (2) 1-shot ICL is challenging due to the limited feedback provided to the LLM about how to improve the user prompt, and thus only minor improvements in prompt-image consistency can be obtained over random paraphrasing.

Table 10: Relative prompt-image consistency improvement between the user prompt and the best prompt found, averaged across prompts.

| Method | LLM | T2I | MSCOCO | | P2 | |
|---|---|---|---|---|---|---|
| | | | $\mathcal{S}_{\text{DSG}}(p_0, g(p_0)) < 1$ | All | $\mathcal{S}_{\text{DSG}}(p_0, g(p_0)) < 1$ | All |
| Paraphrasing | Llama-2 | LDM-2.1 | +17.74 | +10.67 | +21.95 | +19.22 |
| OPT2I | | | **+18.63** | **+11.21** | **+25.39** | **+22.24** |
| Paraphrasing | GPT-3.5 | LDM-2.1 | +17.52 | +10.53 | +21.80 | +19.09 |
| OPT2I | | | **+18.05** | **+10.85** | **+22.58** | **+19.77** |
| Paraphrasing | Llama-2 | CDM-M | +18.65 | +10.53 | +26.54 | +22.24 |
| OPT2I | | | **+19.61** | **+11.07** | **+29.19** | **+24.93** |

## B.2    Filtering out already consistent user prompts

When adopting the DSG objective, user prompts might already achieve a perfect consistency score initially, *i.e.*, $\mathcal{S}_{\text{DSG}}(p_0, g(p_0)) = 1$. We observe this phenomenon happens with higher frequency on the simpler

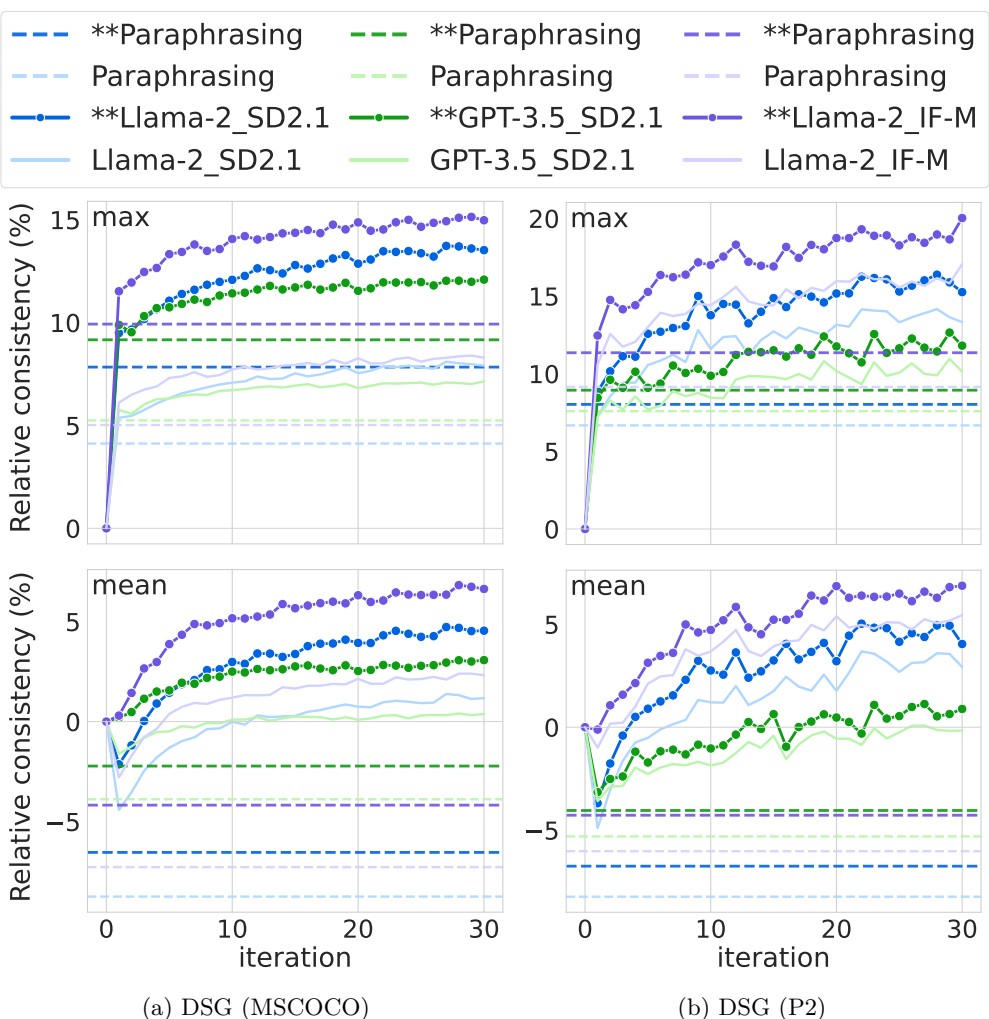

Figure 7: OPT2I optimization curves obtained with prompts having $\mathcal{S}_{\text{DSG}}(p_0) < 1$, marked by "**" and full colors. In contrast, faded-color curves consider all prompts. Each plot tracks either the *max* or the *mean* relative improvement in consistency across revised prompts per iteration.

MSCOCO prompts ($\sim 40\%$), and with less frequency on the more complex PartiPrompts prompts ($\sim 10\%$). Since $\mathcal{S}_{\text{DSG}}(p_0, g(p_0))$ can be computed beforehand, we can avoid optimizing for those user prompts that have already a perfect initial $\mathcal{S}_{\text{DSG}}$ and better showcase the optimization performance of OPT2I. We provide the updated optimization curves in Figure 7, and report the final results in Table 10. In both cases, we highlight results obtained by filtering out "perfect" user prompts with full colors, and contrast them against results obtained with all prompts in faded colors (equivalent to Figure 3).

In Table 10, we observe a higher relative improvement for both MSCOCO and PartiPrompts in all configurations when filtering out "perfect" user prompts, which is more prominent for MSCOCO because the number of excluded prompts is higher. In Figure 7, we observe similar consistent and considerable increases of all optimization curves when considering both *mean* and *max* consistency improvement. In the *mean* case, we remark a reduction in the initial dip in relative consistency, especially in MSCOCO, where OPT2I reaches a positive relative consistency much earlier, *i.e.*, it = $[6, 2, 2]$ vs. it = $[23, 8, 5]$ with `Llama-2`, `GPT-3.5`, and `CDM-M`, respectively.

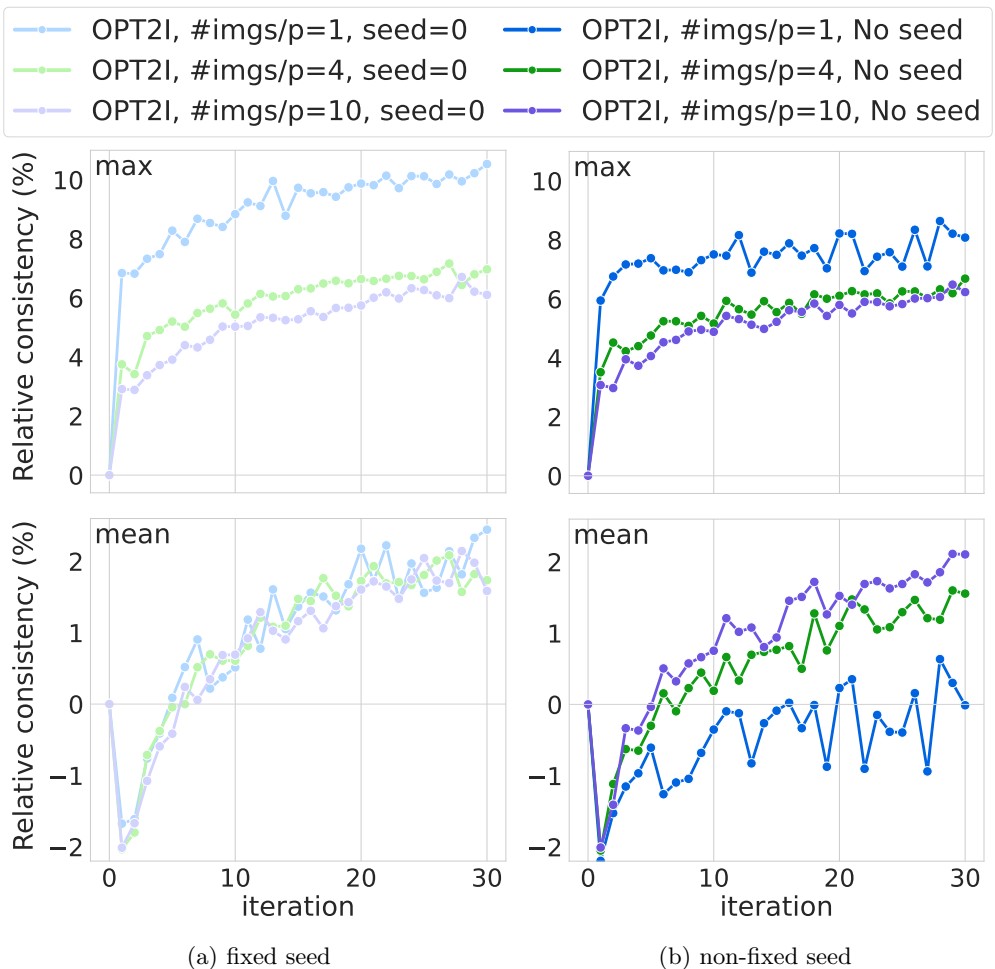

Figure 8: OPT2I optimization curves with (a) fixed or (b) non-fixed seed. Each curve uses optimizes a different number of images per prompt. Y-axis is aligned between (a) and (b). Curves obtained with `Llama-2` and `LDM-2.1` on 200 out of the 2000 prompts from MSCOCO.

### B.3   Impact of seed-fixing and #images/prompt

In this experiment, we ablate the impact of fixing the random seed of the initial noise for the diffusion model throughout the optimization process when optimizing different numbers of images/prompt. We use our default configuration with `Llama-2` and `LDM-2.1` on MSCOCO. In Figure 8a, we show the optimization curves obtained when optimizing 1, 4 (default), and 10 images/prompt with fixed image seed. As expected, we observe no meaningful differences in *mean* consistency improvement. In contrast, the *max* consistency improvement shows a clear distinction between optimizing a single image (single seed) and optimizing 4 or 10, with the former achieving more substantial improvements. We argue that when optimizing a single image seed, OPT2I is more sensitive to changes in the prompts, *i.e.*, there is a higher variance among the scores of revised prompts. We then contrast the optimization curves with fixed seed (8a) against the non-fixed seed ones (8b). Our hypothesis is that optimizing, when not fixing the seed, generating too few images/prompt leads to an unstable/unreliable feedback for the LLM due to the high variance of the generations. Indeed, looking at the optimization curves, we notice that optimizing a single image without fixing the seed is more difficult for OPT2I, which results in a noisy and less steep trajectory, especially in the *mean* case. In contrast, when OPT2I optimizes 4 or 10 images/prompt with no fixed seed, both the *max* and *mean* curve remain similar w.r.t. to using a fixed seed. This supports our choice of generating 4 images/prompt, as it provides

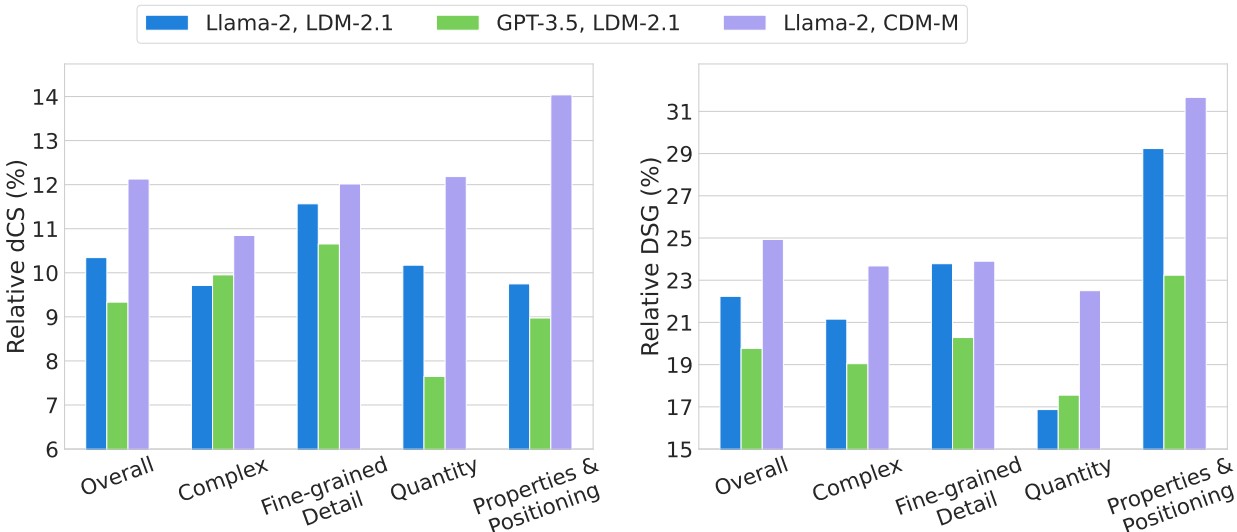

Figure 9: Relative improvement in prompt-image consistency between the user prompt and the best prompt found, averaged across PartiPrompts prompts and broken down by challenge aspect.

enough diversity in the generations while being substantially more computationally efficient than generating 10.

## B.4 Stratified PartiPrompts results

Figure 9 shows the relative improvement in consistency score (dCS or DSG) on prompts from PartiPrompts (P2), broken down by challenge aspect. Note that we only sampled prompts from four of the most difficult dimensions in P2: "Complex", "Fine-grained Detail", "Quantity", and "Properties & Positioning". Intuitively, this plot shows what kinds of prompts are easier to optimize for OPT2I when using different LLMs, T2I models and consistency scores.

The most significant improvement in consistency is observed for prompts related to "Properties & Positioning" when using `Llama-2` in conjunction with `CDM-M` and dCS. Similarly, the combination of `Llama-2`, `CDM-M`, and DSG yields the best results for prompts about "Quantity". For other challenges, `CDM-M` continues to provide the most substantial consistency improvement, although the margin is narrower compared to `LDM-2.1`. Interestingly, `GPT-3.5` shows the smallest improvement in consistency for prompts about "Quantity", regardless of whether dCS or DGS metrics are used. Consistency improvements for prompts from the "Complex" and "Fine-grained Detail" challenges are comparable, which is expected due to their inherent similarities.

## B.5 Why is `GPT-3.5` not as good as `Llama-2`?

In Figure 3 and Table 1, we observe that OPT2I achieves worse results when using `GPT-3.5` as the LLM. Notably, the optimization curves with `GPT-3.5` are flatter than when using `Llama-2`. This result is rather surprising, as current leaderboards (Chiang et al., 2024) indicate that `GPT-3.5` generally outperforms `Llama-2` on a wide variety of NLP tasks. So, in this experiment, we aim to shed light on the possible causes. Given the closed (and expensive) access to `GPT-3.5`, our initial exploration of the meta-prompt structure and phrasing was based on `Llama-2`, and later on we used the exact same prompt with `GPT-3.5`. Hence, one hypothesis for the observed phenomenon is that our meta-prompt is better optimized for `Llama-2`. Another hypothesis is that each LLM has a different balance point between exploration and exploitation for the same sampling temperature of 1.0. In particular, given the flatter optimization curves drawn by `GPT-3.5`, we conjecture that it explores less diverse prompts than `Llama-2`. To verify this, we analyze some text properties of the revised prompts generated by both LLMs.

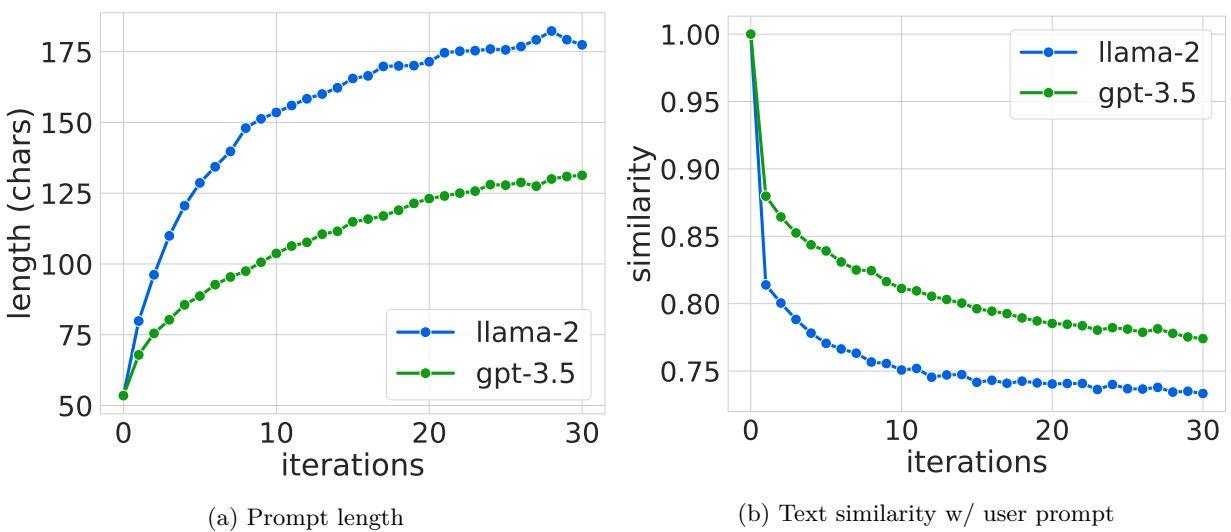

(a) Prompt length

(b) Text similarity w/ user prompt

Figure 10: Text analysis of revised prompts generated by `Llama-2` and `GPT-3.5`.

Figure 10a tracks the length (in number of characters) of the revised prompts at each iteration, and Figure 10b tracks CLIP text similarity between revised prompts and the user prompt along the optimization process, both averaged over the revised prompts generated at the same iterations and over all prompts in the dataset. We observe that when using `Llama-2` for OPT2I, the revised prompts generated at each iteration are longer and more semantically dissimilar to the user prompt compared to those generated by `GPT-3.5`. This means that OPT2I benefits from greater prompt diversity to find the best T2I prompts that lead to more consistent images, which is better achieved with `Llama-2`. Additionally, we note that both the prompt length and the semantic similarity with the user prompt start plateauing around the maximum number of iterations we set, which further validates our selected value of 30. We leave as future work ablating for the sampling temperature with both LLMs.

### B.6 Additional qualitative examples

Figures 11 and 12 show some additional selected examples of user prompt and revised prompts throughout the optimization process, along with the generated images and consistency scores. In particular, we select revised prompts such that the consistency score of the generated images (w.r.t. the user prompt) is strictly higher than the previous best score found so far, *i.e.*, the *leaps* in prompt-image consistency.

Figure 11 shows revised prompts generated with DSG as the scorer. Since DSG is computed as an average of binary scores, it is more coarse than CLIPScore and thus there are fewer leaps in consistency. Overall, we observe that the intermediate revised prompt manages to increase the consistency score in some of the generated images but not for all of them. The best prompt, however, usually manages to improve all 4 generated images.

Figure 12 shows revised prompts generated with dCS as the scorer. In this case, we can see a gradual increase in average dCS, which visually translates to generated images which are more consistent with the user prompt on average. The strong effect of the initial latent noise in the image structure is evident, yet substantial modifications in the format of the input prompt used to condition the generation lead to significant changes in how the T2I model interprets the structure determined by the initial noise (*e.g.*, between rows 2-3 and 4-5 in the squirrel example). We also note that dCS (CLIPScore averaged over subprompts) can occasionally fall short as an evaluation metric for image-text consistency. This is primarily because dCS tends to focus on the presence of visual elements, overlooking other aspects such as spatial relationships. In the toilet example, for instance, we observe how the generated images progressively become more consistent up to a certain point (around row 6). Beyond this point, the revised prompts and the generated images start degenerating (*e.g.*, by overemphasizing certain elements), while dCS continues to improve. This highlights that, while dCS may

serve as a useful fine-grained consistency metric to provide visual feedback for T2I prompt optimization with an LLM, it may not be as effective for evaluation purposes.

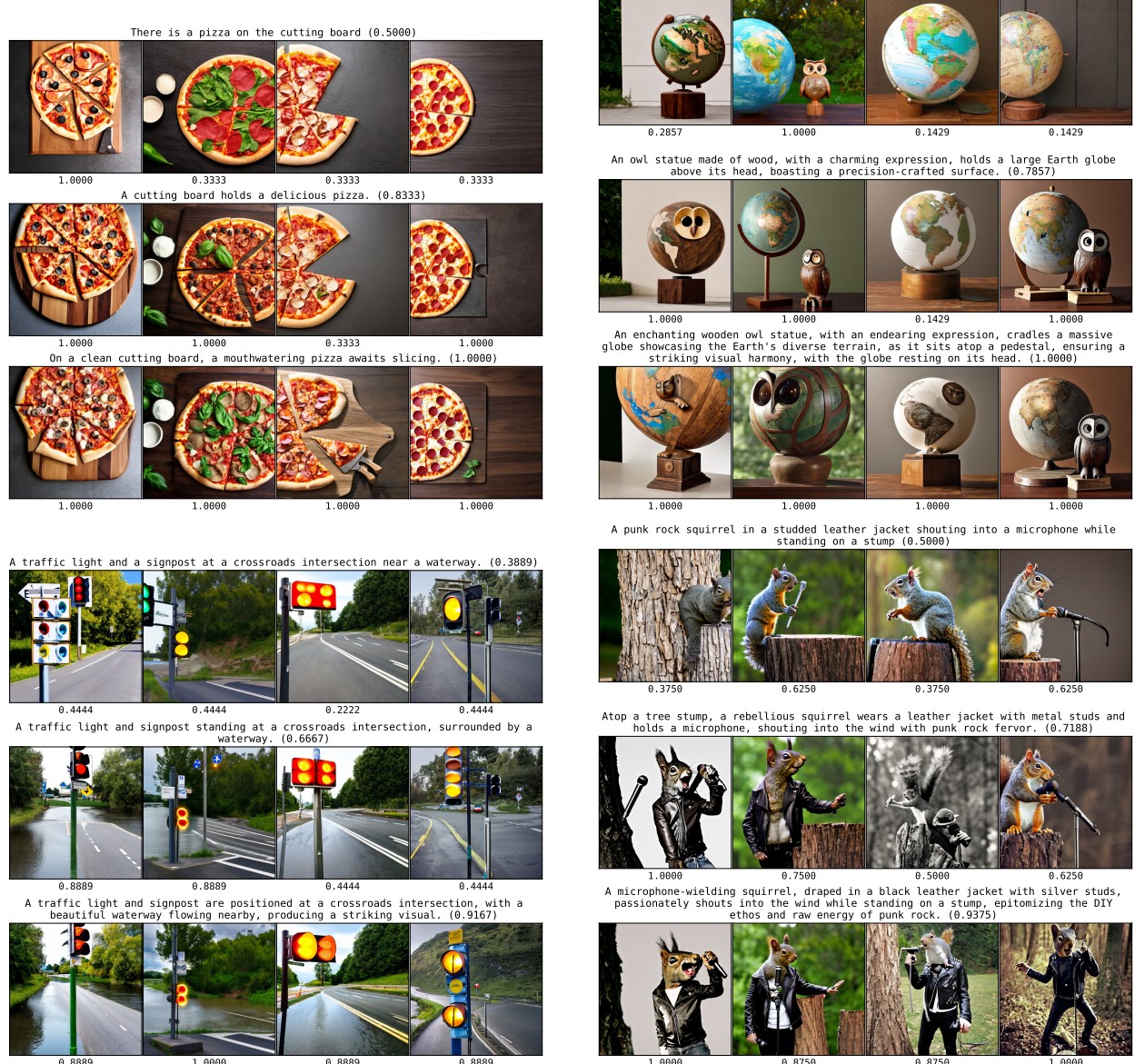

Figure 11: Selected examples of initial prompts from MSCOCO (left) and PartiPrompts (right) and revised prompts across the optimization, along with the generated images. The optimizer refines prompts for `LDM-2.1` using `Llama-2` as LLM and DSG as scorer. We report DSG score averaged across images.

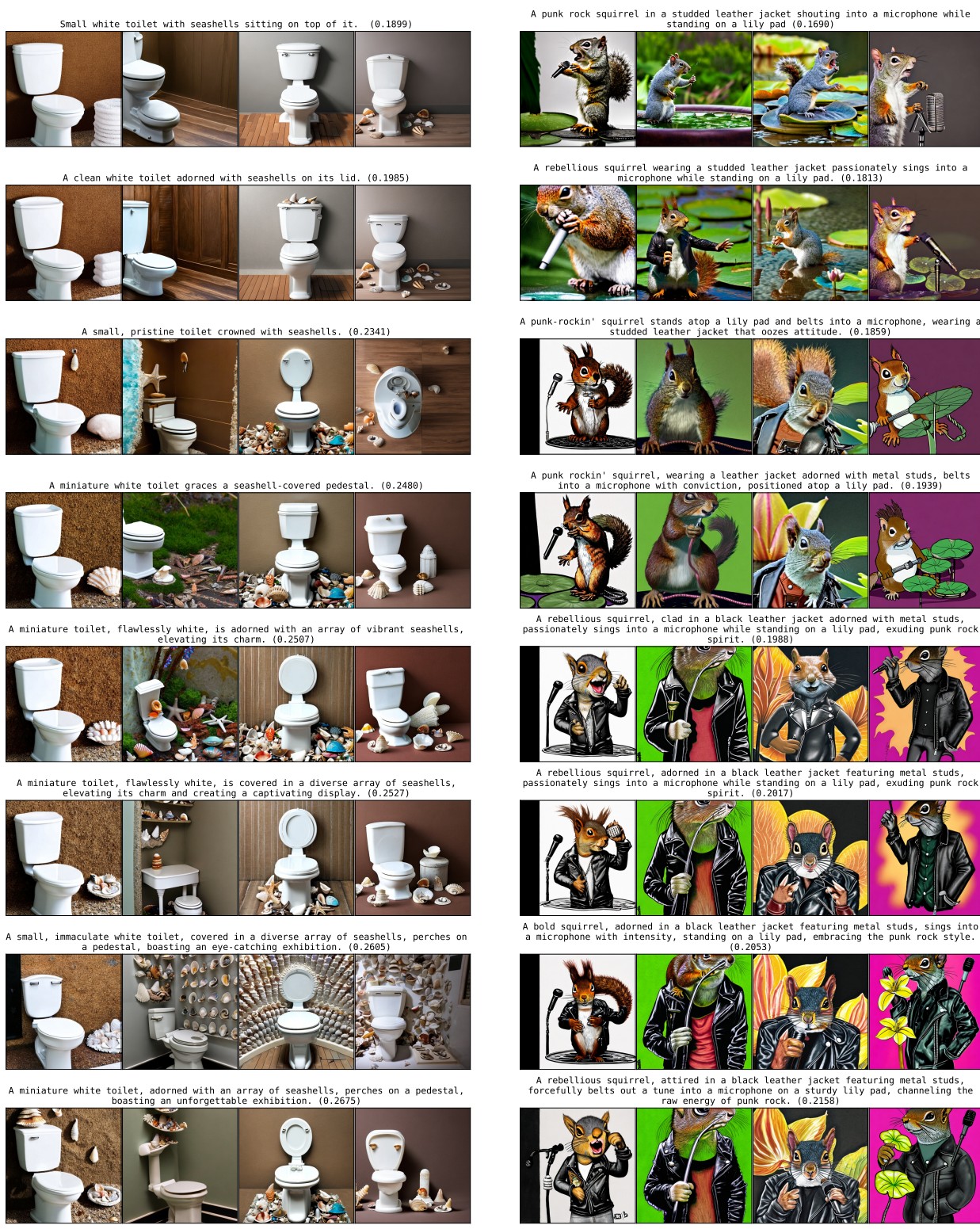

Figure 12: Selected examples of initial prompts from MSCOCO (left) and PartiPrompts (right) and revised prompts across the optimization, along with the generated images. The optimizer refines prompts for `LDM-2.1` using `Llama-2` as LLM and dCS as scorer. We report dCS score averaged across images.

## B.7 Results with updated models

In this section, we report results with more recent (and better) T2I models and LLMs. We updated the T2I model to LDM-XL-Turbo (Sauer et al., 2023) and LDM-3 (Esser et al., 2024) (which is based on a diffusion transformer), and the LLM to the instruction-finetuned Llama-3.1 (8B and 70B) (Dubey et al., 2024). In addition, we have upgraded the VQA model that powers the DSG metric to PaliGemma-224 (3B) (Beyer et al., 2024) finetuned on VQAv2 (Goyal et al., 2017).

**Optimization curves and comparison with baselines** We report the results in Figure 13(a-b) and Table 11. Our results show that, while the initial consistency score is higher, OPT2I still manages to considerably improve consistency. For instance, optimizing PartiPrompts' prompts for 30 iterations generating 5 solutions per iteration, Llama-3.1 (70B) with LDM-3 has an initial average DSG score of 84.7%; OPT2I is able to find a rephrase of the initial prompts achieving an average DSG score of 98.8%, yielding a total absolute improvement of 14%. Note that the final DSG score is already very close to 100%, so there's not much more room for improvement. Meanwhile, a (compute-matched) paraphrasing baseline using the same models only achieves an average DSG score of around 97.7%. For LDM-XL-Turbo, the initial DSG score is 72.8% and OPT2I is able to find prompt rephrases with an average DSG score of 93.7% (absolute improvement of 20.9%), while the paraphrasing baseline is only able to improve up to 91.5%. In every case, OPT2I consistently outperforms the paraphrasing baseline, which is in line with our results for older models (see first two rows of Table 11). Therefore, we conclude that our framework increases the consistency even for more recent T2I models.

**Success rate** In Figure 13(c), we report the success rate throughout the optimization process. At a given iteration, we define success rate as the percentage of generated paraphrases that achieve a strictly higher consistency score than the initial prompt. Because of this, here we only consider prompts that can be improved, *i.e.*, that do not already have an initial DSG score of 100%. We observe that, when optimizing prompts with OPT2I, the success rate surpasses 50% after 10 iterations, reaching up to 60-70% after 30 iterations. Instead, the paraphrasing baseline only achieves a score between 30-40%.

Table 11: Absolute (DSG) and relative ($\Delta$ DSG) improvement in T2I consistency between the user prompt and the best prompt found, averaged across all prompts in the dataset.

| Method | LLM | T2I | DSG | $\Delta$ DSG |
|---|---|---|---|---|
| Paraphrasing | Llama-2 (70B) | CDM-M | 86.32 | +22.24 |
| OPT2I | | | 88.48 | **+24.93** |
| Paraphrasing | Llama-3.1 (8B) | LDM-XL-Turbo | 90.51 | +17.68 |
| OPT2I | | | **93.02** | **+20.18** |
| Paraphrasing | Llama-3.1 (70B) | LDM-XL-Turbo | 91.50 | +18.70 |
| OPT2I | | | **93.68** | **+20.85** |
| Paraphrasing | Llama-3.1 (70B) | LDM-3 | 97.67 | +12.99 |
| OPT2I | | | **99.04** | **+14.10** |

## B.8 Complete optimization example

Here we provide a detailed example list of prompt paraphrases along with their fine-grained DSG scores. We consider the user prompt `"A dignified beaver wearing glasses, a vest, and colorful neck tie. He stands next to a tall stack of books in a library."` (from PartiPrompts). We use `Llama-3.1 (70B)`, `LDM-XL-Turbo` and DSG-PaliGemma. We generate 3 new prompts per iteration, 4 images per prompt, and we early-stop when a perfect score of 100% is reached, which takes 7 iterations and 2 minutes 12 seconds in this case. In Figures 14/ 15, we show a prompt paraphrase from each iteration (including the initial prompt at iteration 0) and its corresponding DSG score, including the evaluation questions and the average VQA scores for the 4 images.

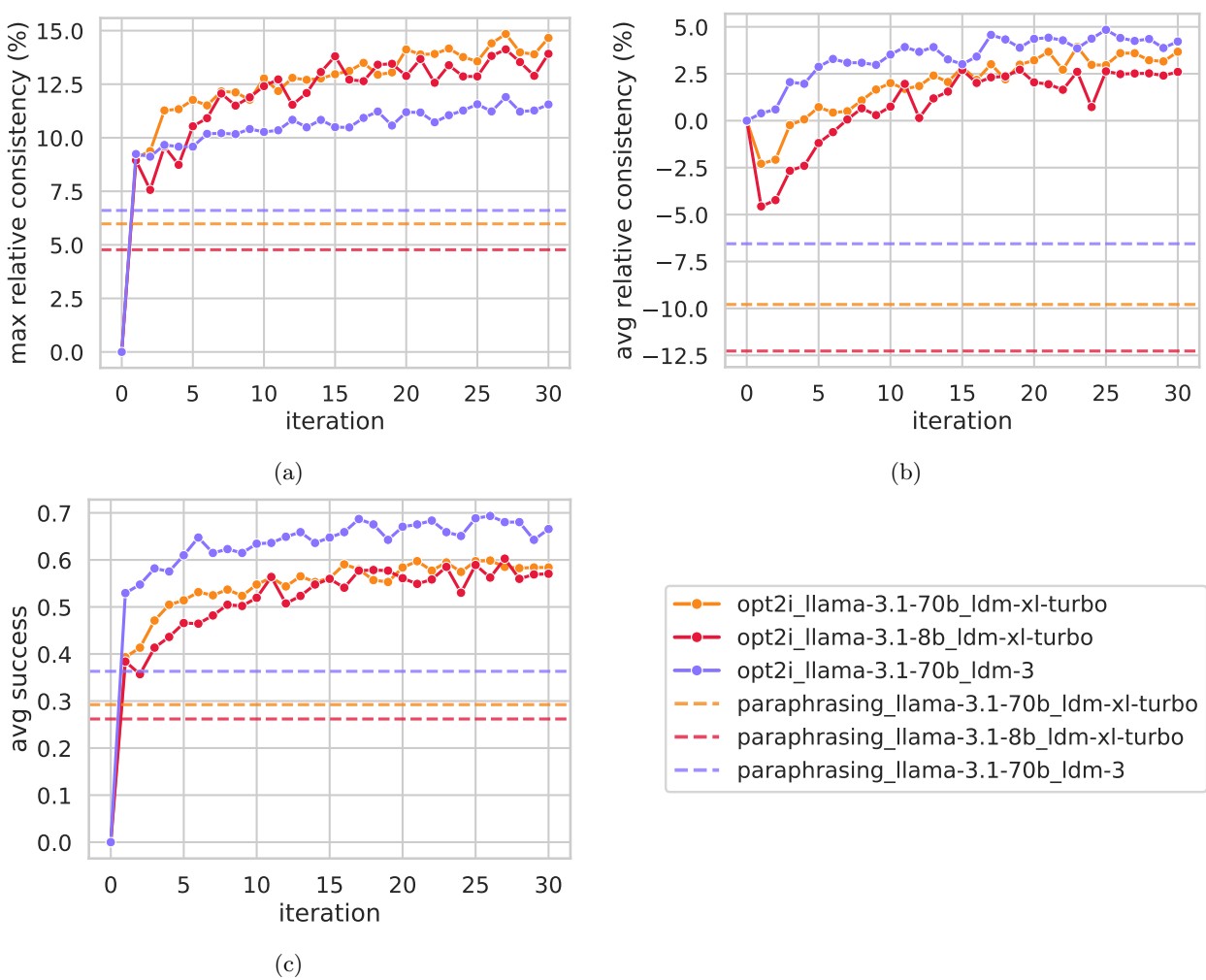

Figure 13: OPT2I curves with updated T2I models and LLMs. We report maximum (a) and average (b) DSG scores, and also average success rate (c) across the revised prompts at each iteration.

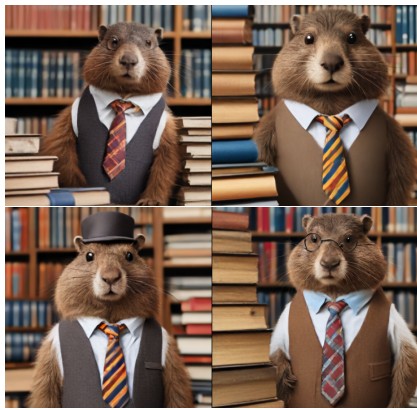

0. "A dignified beaver wearing glasses, a vest, and colorful neck tie. He stands next to a tall stack of books in a library."

```
overall score:  90
evaluation questions:
Is there a beaver?  100
Does the beaver have glasses?  25
Does the beaver have a vest?  100
Does the beaver have a neck tie?  100
Is the beaver dignified?  100
Is the neck tie colorful?  100
Is the beaver standing?  75
Are there books?  100
Are the books in a tall stack?  100
Is there a library?  100
Is the beaver next to the books?  75
Is the beaver in the library?  100
Are the books in the library?  100
```

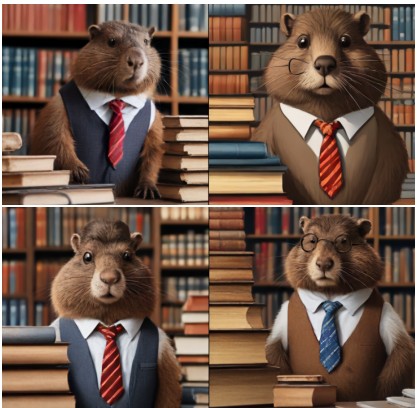

1. "A dignified beaver, wearing glasses perched on the end of its nose, a vest, and a vibrant neck tie, stands upright next to a tall stack of books in a quiet library."

```
overall score:  86
evaluation questions:
Is there a beaver?  100
Does the beaver have glasses?  50
Does the beaver have a vest?  75
Does the beaver have a neck tie?  100
Is the beaver dignified?  100
Is the neck tie colorful?  100
Is the beaver standing?  25
Are there books?  100
Are the books in a tall stack?  75
Is there a library?  100
Is the beaver next to the books?  100
Is the beaver in the library?  100
Are the books in the library?  100
```

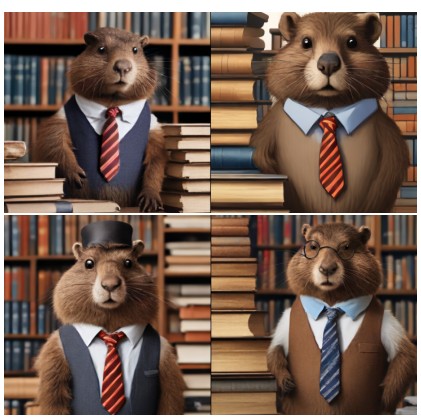

2. "A dignified beaver, wearing glasses perched on its nose, a vest, and a vibrant neck tie, stands upright with confidence beside a tall stack of books neatly arranged within arm's reach on a library bookshelf."

```
overall score:  88
evaluation questions:
Is there a beaver?  100
Does the beaver have glasses?  25
Does the beaver have a vest?  75
Does the beaver have a neck tie?  100
Is the beaver dignified?  100
Is the neck tie colorful?  75
Is the beaver standing?  100
Are there books?  100
Are the books in a tall stack?  100
Is there a library?  100
Is the beaver next to the books?  75
```

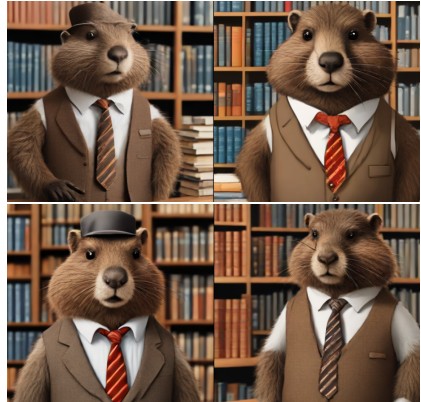

3. "With poise and elegance, a well-dressed beaver stands directly adjacent to a high-rise bookshelf in a library, wearing glasses that sit comfortably on its nose, along with a neat vest and a bright, eye-catching neck tie."

```
overall score:  86
evaluation questions:
Is there a beaver?  100
Does the beaver have glasses?  50
Does the beaver have a vest?  100
Does the beaver have a neck tie?  100
Is the beaver dignified?  100
Is the neck tie colorful?  75
Is the beaver standing?  75
Are there books?  100
Are the books in a tall stack?  75
Is there a library?  100
Is the beaver next to the books?  50
Is the beaver in the library?  100
Are the books in the library?  100
```

Figure 14: Example of complete optimization. Part 1/2

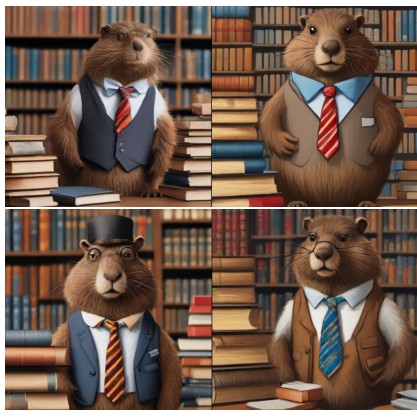

4. "In a library, a refined beaver stands proudly adjacent to a tall, orderly stack of books, wearing a pair of glasses, a vest, and a colorful neck tie, exuding an air of dignity."

```
overall score:  94
evaluation questions:
Is there a beaver?  100
Does the beaver have glasses?  50
Does the beaver have a vest?  100
Does the beaver have a neck tie?  100
Is the beaver dignified?  100
Is the neck tie colorful?  100
Is the beaver standing?  75
Are there books?  100
Are the books in a tall stack?  100
Is there a library?  100
Is the beaver next to the books?  100
Is the beaver in the library?  100
Are the books in the library?  100
```

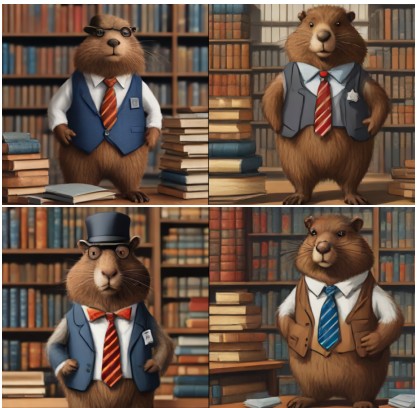

5. "In a library, a refined beaver stands confidently on its hind legs, showcasing its dignified demeanor, while wearing glasses that sit comfortably on the bridge of its nose, a vest, and a colorful neck tie, next to a towering stack of books."

```
overall score:  94
evaluation questions:
Is there a beaver?  100
Does the beaver have glasses?  50
Does the beaver have a vest?  100
Does the beaver have a neck tie?  100
Is the beaver dignified?  100
Is the neck tie colorful?  100
Is the beaver standing?  100
Are there books?  100
Are the books in a tall stack?  100
Is there a library?  100
Is the beaver next to the books?  75
Is the beaver in the library?  100
Are the books in the library?  100
```

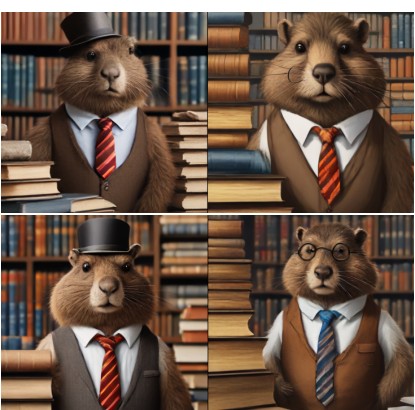

6. "A dignified beaver, adorned with a pair of spectacles resting comfortably on its nose, stands upright beside a towering stack of books in a library, dressed in a vest and a vibrant neck tie, exuding an air of refinement."

```
overall score:  94
evaluation questions:
Is there a beaver?  100
Does the beaver have glasses?  50
Does the beaver have a vest?  100
Does the beaver have a neck tie?  100
Is the beaver dignified?  100
Is the neck tie colorful?  100
Is the beaver standing?  75
Are there books?  100
Are the books in a tall stack?  100
Is there a library?  100
Is the beaver next to the books?  100
Is the beaver in the library?  100
Are the books in the library?  100
```

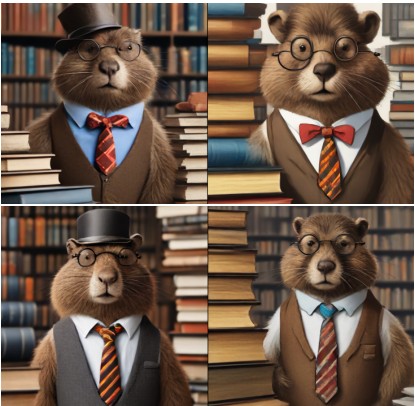

7. "A dignified beaver, sporting a pair of wire-rimmed glasses with lenses that sparkle in the library light, stands proudly next to a towering stack of books, adorned with a colorful neck tie and a vest that complements its air of sophistication."

```
overall score:  100
evaluation questions:
Is there a beaver?  100
Does the beaver have glasses?  100
Does the beaver have a vest?  100
Does the beaver have a neck tie?  100
Is the beaver dignified?  100
Is the neck tie colorful?  100
Is the beaver standing?  100
Are there books?  100
Are the books in a tall stack?  100
Is there a library?  100
Is the beaver next to the books?  100
Is the beaver in the library?  100
Are the books in the library?  100
```

Figure 15: Example of complete optimization. Part 2/2

