# OpenReview forum: "Improving Text-to-Image Consistency via Automatic Prompt Optimization"
_TMLR — Accepted by TMLR_

### Review · Reviewer_X6VC · 2024-07-15

**Summary Of Contributions:**

This paper studies text-to-image generative models. Aiming to improve prompt-image consistency, they propose a novel framework, OPT2I, using LLM for such improvement. The method is training-free and expected to be generalizable to any T2I models. The authors claimed that the experiment results show the effectiveness of the proposed method.

**Audience:**

Yes

**Broader Impact Concerns:**

There are no such concerns.

**Claims And Evidence:**

Yes

**Requested Changes:**

- I would like to see the successful rate and/or standard variance of the quantitative results to understand how robust and likely the LLM can improve the results.
- If possible, I would like to see the results on more T2I models, e.g., SD1.5, SDXL, SD3.
- Some non-T2I generative models, like Instruct-Pix2Pix, which is an image editing model and is IT2I (image+text to image), can also be regarded as T2I when the image is fixed.
    - i.e., the original model can be defined as$ f(I, t) \to I'$; for a given image $I_1$, let $g(t) = f(I_1, t)$, then $g(t)$ can be regarded as a T2I model.
    - I wonder if the proposed model can also work in this scenario to optimize the best editing prompt. Successfully achieving this will make the model more general and powerful.

**Strengths And Weaknesses:**

### Strengths
- The method is training-free and independent of specific T2I images.
- The method is well-motivated and described in detail (e.g., Sec.2.3. Optimization objective)
- Extensive quantitative and qualitative results are provided to support the claims.


### Weaknesses
- The DSG questions and dCS decompositions are also generated by the LLM. This may lead to a situation: if the LLM does not notice or care about some points of the generation's requirement, it will always be ignored in the following optimization processes.
- The "optimization" of the desired metrics is not tailored. It depends on the decision of LLM and luck to achieve successful editing.
- The tested T2I models cannot perfectly support the claims.
    - The T2I models are not the state-of-the-art at this point. For example, the most powerful version of Stable Diffusion is SDXL or SD3, which has a larger capacity and more powerful text encoders
    - On the other hand, only considering state-of-the-art T2I models may lead to incomplete conclusions. For example, a weaker model like SD1.5 may have a lower ability to reason the text prompts and, therefore, a more challenging process to optimize through the proposed framework.
    - (Minor) The LLMs are also not state-of-the-art. This is not a problem since a weaker LLM can better support the claims.
- (Minor) The proposed model has some efficiency issues.

---

> ### Author Response · Authors · 2024-09-25
> **Response to Reviewer X6VC**
>
> We thank Reviewer X6VC for their thoughtful comments! Below we respond to individual questions/concerns.
>
> ***“If the LLM used in DSG/dCS misses some elements from the initial prompt, this will harm the optimization process.”***
> As explained in the limitations section, the performance of our method hinges on the quality of prompt-image consistency scores, which we assume to be *good enough*. As shown in Figures 11 and 12, DSG/dCS may occasionally make evaluation mistakes. However, the optimization curves depicted in Figures 3 and 13 the final results reported in Tables 1 and 11 demonstrate that these metrics, in their current state, provide enough signal to guide the optimization process.
>
> ***“The optimization of the metrics depends on the LLM’s decision and luck.”***
> A contribution of our work is showing to what extent an LLM can be used to optimize prompts for T2I models. We note that, within our framework, prompt rephrases are not random or due to “luck” since the LLM is informed about previous successful prompt rephrases and their corresponding fine-grained consistency scores, which are key in guiding the optimization process. We compare OPT2I with a paraphrasing baseline in which prompt rephrases are generated randomly without any context, and show that the additional context helps in reaching higher consistency scores.
>
> ***“The proposed model has some efficiency issues.”***
> We refer X6VC to our limitations section for a discussion on this point. Moreover, with updated smaller/faster models (LDM-XL-Turbo, DSG-PaliGemma and Llama-3.1 (70B)) and more recent GPUs (A100s), running the optimization for 10 iterations (generating 5 prompt paraphrases per iteration and 4 images per prompt) takes \~2.5 minutes vs. \~27 minutes previously (with LDM-2.1, DSG-InstructBLIP, Llama-2 (70B) and V100 GPUs).
>
> ***“Missing success rate or standard variance of quantitative results.”***
> In Figure 13(c), we report the success rate throughout the optimization process. At a given iteration, we define success rate as the percentage of generated paraphrases that achieve a strictly higher consistency score than the initial prompt. Because of this, here we only consider prompts that can be improved, i.e., that do not already have an initial DSG score of 100%. We observe that, when optimizing prompts with OPT2I, the success rate surpasses 50% after 10 iterations, reaching up to 60-70% after 30 iterations. Instead, the paraphrasing baseline only achieves a score between 30-40%.
>
> ***“Can OPT2I be extended to optimize image editing prompts?”***
> We are excited at the possibility of extending our framework to optimize image editing prompts, and believe it should work. The main challenge would be finding a good scoring function to evaluate edited images. However, this application is out of the scope of our paper and we leave it for future work.

---

### Review · Reviewer_GmPJ · 2024-07-30

**Summary Of Contributions:**

The authors propose OPT2I, a training-free optimization-by-prompting framework designed to enhance prompt-image consistency in Text-to-Image (T2I) models. The proposed method refines prompts without requiring any parameter updates, making it versatile and compatible with various T2I models, Large Language Models (LLMs), and consistency metrics. By leveraging a pre-trained T2I model, an LLM, and an automatic prompt-image consistency score (e.g., CLIPScore or DSG), OPT2I iteratively improves user-provided prompts by suggesting alternatives that lead to more consistent images. Extensive experiments demonstrate the effectiveness of OPT2I on the MSCOCO dataset and 24.9% on the PartiPrompts while preserving the Fréchet Inception Distance (FID).

**Audience:**

Yes

**Claims And Evidence:**

Yes

**Requested Changes:**

More analysis of the DSG and potential limitations of the current framework is necessary. Wether the dCS is rational remains unclear. Additional experiments on more text-to-image models can strengthen the paper.

**Strengths And Weaknesses:**

Strengths:

1: The introduced OPT2I is a training-free framework, making it easy to transfer to other advanced text-to-image models and LLMs.

2: The newly proposed metric is effective in evaluating the consistency between input text and generated images.

3: Text-based in-context learning for improving consistency is interesting and heavily reduces the trivial attempts on prompt design, which will be generally used practically.

Weaknesses:

1: The rationality of decomposed CLIPScore (dCS) to measure the consistency. As the shown examples in this paper, the generated images always contain **multiple objects** and the CLIPScore is originally used to measure the global or overall similarity between text and image. Therefore, with the decomposed text like "a bike," if we compute the similarity between this decomposed text and the whole image, can it reflect the consistency used for meta-prompt construction? If so, could the authors provide more illustrations?

2: Experiments with more text-to-image generation models. Besides the used latent diffusion model (LDM), will the proposed framework be in other generation architectures such as text-to-image diffusion transformers or API-based DALLE?

3: The question design/generation in DSG or potential measurement limitations in the current consistency metric. From the 3rd image in the 4th row of Figure 4(d), the text contains the description "**four** teacups," and the generated images may have **five** or **three**, which means the inconsistency still exists. Meanwhile, the consistency scores are 1.0. Could the authors provide more analysis of similar situations or analyze the current limitations of the proposed metric for consistency evaluation?

4: (Not a weakness but more like a suggestion or discussion) More detailed analysis of potential sub-metrics for different failure modes. The authors emphasize several failure modes in the Introduction, such as missing objects, the wrong amount of objects, object attribute errors, and non-compliance. I haven't seen the metrics for each mentioned mode to evaluate the text-to-image generation. Can the proposed framework obtain apparent performance gain on these specific modes (with corresponding metrics)?

Minor:

The sentence "a plethora of" in the abstract. This sounds like it has an excessive quantity.

---

> ### Author Response · Authors · 2024-09-25
> **Response to Reviewer GmPJ**
>
> We thank Reviewer GmPJ for their thoughtful comments! Below we respond to individual questions/concerns.
>
> ***“Is CLIPScore sensible when computing the similarity between a single object and the whole image?”***
> One of the tasks CLIP was evaluated on in the original paper \[1\] is zero-shot image classification. In that setting, the text also contains a single object and is compared against the whole image. Another example is Hall et al. \[2\], where the text prompt also contains a single object. Given this precedent, we conclude CLIP is sensible enough to be used in dCS.
>
> ***“Deeper analysis on the limitations of DSG score?”***
> We refer GmPJ to section 4.2 of the DSG score paper \[3\] for such an analysis.
>
> ***“Deeper analysis of sub-metrics for different failure modes? e.g., wrong amount of objects, object attribute errors, general non-compliance”***
> We refer GmPJ to Appendix B.4 and Figure 9 of our paper, where we break down consistency scores by aspect, as defined by PartiPrompts: “complex”, “fine-grained detail”, “quantity” and “properties & positioning”. “Wrong amount of objects” would fall into the “quantity” category, object attribute errors would fall into the “properties & positioning” category, and general non-compliance would fall into the “complex” and “fine-grained detail” categories.
>
> \[1\] Radford, Alec, et al. "Learning transferable visual models from natural language supervision." International conference on machine learning. PMLR, 2021\.
> \[2\] Hall, Melissa, et al. "DIG In: Evaluating Disparities in Image Generations with Indicators for Geographic Diversity." Transactions on Machine Learning Research.
> \[3\] Cho, Jaemin, et al. "Davidsonian Scene Graph: Improving Reliability in Fine-grained Evaluation for Text-to-Image Generation." The Twelfth International Conference on Learning Representations.

---

### Review · Reviewer_3vv4 · 2024-09-11

**Summary Of Contributions:**

This paper leverages an external Larage Language Model (LLM) to provide prompt-image consistency feedback and paraphrase the input prompt to improve the synthesized image from the target text-to-image model (T2I). Through this iterative optimization, the proposed OPT2I can significantly boost the consistency score yet maintain a competitive FID score.

**Audience:**

Yes

**Broader Impact Concerns:**

As the used LLMs may produce hallucinations when paraphrasing the prompt, this error may propagate to the final T2I result.

**Claims And Evidence:**

Yes

**Requested Changes:**

1. Add a discussion to compare the differences with previous works on this similar prompt paraphrasing method.
2. Add a discussion of prompt paraphrasing examples and their quality.
3. Add experiments of more/excessive paraphrasing iterations.

**Strengths And Weaknesses:**

**Strengths**
+ The idea of improving prompt-image consistency is well-motivated, where the proposed iterative optimization is reasonable for this scenario.
+ They provide comprehensive ablation studies and qualitative visualization, as well as the used prompts in the system, which is valuable to future research in this community.

**Weaknesses**
- The idea of prompt paraphrasing to improve T2I is not new. There are already several works [1, 2] studying this idea, where [3] also investigate the iterative paraphrasing.
- Though Figure 2 can show the overview of OPT2I, it still needs to provide a detailed process for the entire pipeline—for example, a complete example list of prompt paraphrasings with the related DSG scores.
- According to Figure 10, with more iterations, the prompts are getting longer but less similar to the original ones. Will the leveraged LLM produce hallucinations as well? How will it hurt the performance? How can such error propagations to the end T2I be alleviated?
- According to Figure 7, more iterations can keep leading to better final consistency. Will it be converged after even more iterations? Or the performance may drop with too excessive iterations.

**Reference**
- [1] Improving Image Generation with Better Captions
- [2] (NeurIPS'23) Optimizing Prompts for Text-to-Image Generation
- [3] (EMNLP'23) Collaborative Generative AI: Integrating GPT-k for Efficient Editing in Text-to-Image Generation

---

> ### Author Response · Authors · 2024-09-25
> **Response to Reviewer 3vv4**
>
> We thank Reviewer 3vv4 for their thoughtful comments! Below we respond to individual questions/concerns.
>
> ***“The idea of prompt paraphrasing to improve T2I is not new.”***
> OPT2I diverges significantly from existing related works in its unique approach, which involves an iterative optimization process driven by recursive interactions between the T2I model, a scoring function, and an LLM, leading to T2I prompts that effectively maximize the scoring function. Concretely:
> DALLE-3 \[1\] recaptions images and trains a T2I model on synthetic captions. Then, at inference time, it relies on the same LLM used for recaptioning to rewrite/paraphrase user prompts. Note that this method does not generalize to other T2I models without retraining. Instead, OPT2I maintains the T2I model frozen and paraphrases prompts/captions to optimize a consistency score.
> Promptist \[2\] finetunes a language model on a dataset of human edits to T2I prompts with the goal of improving image aesthetics. Ultimately, the language model naively learns to just add modifiers to the original prompt. Note that we directly compare to Promptist in section 3.2, showing that our method achieves higher consistency.
> Zhu et al. \[3\] take a similar approach to Promptist to improve image aesthetics, also requiring a dataset of prompt transformations to adapt a language model via finetuning or in-context learning. Instead, OPT2I targets prompt-image consistency and not aesthetics, paraphrases the whole prompt compared to adding only prompt modifiers/tags, and does not rely on any pre-existing dataset of prompt transformations, and instead builds it on-the-fly throughout the optimization process using an LLM via in-context learning.
>
> ***“Will the LLM produce prompt rephrases with hallucinations?”***
> Based on our qualitative analysis, the rephrased prompts might grow longer because the LLM expresses certain concepts in a more verbose way to increase their influence, or because the LLM adds extra elements that were not present in the original prompt. In this context, adding extra elements would not be considered a hallucination since the initial prompt is underspecified wrt the generated image.
>
> ***“If the optimization process is run for more iterations, will the consistency score plateau or drop?”***
> In general, the consistency score is inherently limited by the capabilities of the underlying T2I model; a strong model which is sensitive to prompt rephrases (e.g. LDM-3) may achieve a near-perfect consistency score, while a weaker model may only attain a lower maximum consistency score. To further investigate this, we extended the optimization process for 20 more iterations (up to 50\) using the Llama-3.1 (70B), LDM-XL-Turbo, DSG-PaliGemma variant. Our results show that, after 50 iterations, max/avg consistency scores already start to plateau. We attribute this –scores plateauing and not dropping– to the fact that the LLM has access to a history of high-scoring prompts when generating new rephrases.
>
> ***“Can you provide a complete example list of prompt paraphrases with their DSG scores?”***
> In Appendix B.8 and Figures 14/15, we provide a detailed example list of prompt paraphrases along with their fine-grained DSG scores at each iteration.

---

### Author Response · Authors · 2024-09-25
**General response**

We thank the reviewers for their thoughtful comments! We are pleased that they found our method well-motivated (X6VC, 3vv4), interesting, and useful (GmPJ). The reviewers also appreciated the flexibility and training-free nature of our approach (X6VC, GmPJ). Additionally, we appreciate X6VC's acknowledgement of our extensive quantitative and qualitative results, which provide strong support for our claims. Furthermore, we are glad that GmPJ believes our comprehensive ablations and visualizations will be beneficial to the research community.

We add the new results in Appendices B.7 and B.8, and respond to general questions/concerns below.

A common concern was whether the performance of our method holds for more recent (and better) T2I models and LLMs. We note that T2I models with better text encoders might be a double-edged sword: they might follow the prompt better initially but also be more sensitive to rephrases.

We have upgraded the T2I model to LDM-XL-Turbo [1] and LDM-3 [2] (which uses a diffusion transformer, as requested by GmPJ), and the LLM to the instruction-finetuned Llama-3.1 (8B and 70B) [3]. In addition, we have upgraded the VQA model that powers the DSG metric to PaliGemma-224 (3B) [4] finetuned on VQAv2. We report the new results in Figure 13 and Table 11.

Our results show that, while the initial consistency score is higher, OPT2I still manages to considerably improve consistency. For instance, optimizing PartiPrompts’ prompts for 30 iterations generating 5 solutions per iteration, Llama-3.1 (70B) with LDM-3 has an initial average DSG score of 84.7%; OPT2I is able to find a rephrase of the initial prompts achieving an average DSG score of 98.8%, yielding a total absolute improvement of 14%. Note that the final DSG score is already very close to 100%, so there’s not much more room for improvement. Meanwhile, a (compute-matched) paraphrasing baseline using the same models only achieves an average DSG score of around 97.7%. For LDM-XL-Turbo, the initial DSG score is 72.8% and OPT2I is able to find prompt rephrases with an average DSG score of 93.7% (absolute improvement of 20.9%), while the paraphrasing baseline is only able to improve up to 91.5%. In every case, OPT2I consistently outperforms the paraphrasing baseline, which is in line with our results for older models (see first two rows of Table 13). Therefore, we conclude that our framework increases the consistency even for more recent T2I models.

[1] Sauer, Axel, et al. "Adversarial diffusion distillation." arXiv preprint arXiv:2311.17042 (2023).
[2] Esser, Patrick, et al. "Scaling rectified flow transformers for high-resolution image synthesis." Forty-first International Conference on Machine Learning. 2024.
[3] Dubey, Abhimanyu, et al. "The llama 3 herd of models." arXiv preprint arXiv:2407.21783 (2024).
[4] Beyer, Lucas, et al. "PaliGemma: A versatile 3B VLM for transfer." arXiv preprint arXiv:2407.07726 (2024).

---

### Decision · Action_Editor_nJpy · 2024-10-15

**Recommendation:** Accept as is

**Comment:**

This paper introduces a new training-free optimization-by-prompting framework designed to enhance prompt-image consistency in Text-to-Image (T2I) models. After author rebuttal, it received 3 Accept recommendations. All the reviewers are happy about the paper, commenting that (1) the model is well-motivated and its performance is verified by extensive experiments; (2) the paper is high quality, and the reply mostly addresses the reviewers' concerns; (3) the pipeline is elegant, easy to follow  and has the potential to benefit general visual generation. Therefore, the Action Editor would also like to recommend acceptance of the paper.

**Audience:**

Yes

**Claims And Evidence:**

Yes